**Variations in surface ozone and carbon monoxide in the Kathmandu Valley and surrounding broader regions during SusKat-ABC field campaign: Role of local and regional sources**

Piyush Bhardwaj[1,2,*], Manish Naja[1], Maheswar Rupakheti[3], Aurelia Lupascu[3], Andrea Mues[3], Arnico K. Panday[4], Rajesh Kumar[5], Khadak Mahata[3], Shyam Lal[6], Harish C. Chandola[2], Mark G. Lawrence[3]

[1]Aryabhatta Research Institute of Observational Sciences (ARIES), Nainital, 263002, India

[2]Dev Singh Bisht Campus, Kumaun University, Nainital, 263001, India

[*]Now at Gwangju Institute of Science and Technology (GIST), Gwangju, 61005, Republic of Korea

[3]Institute for Advanced Sustainability Studies (IASS), Potsdam, 14467, Germany

[4]International Centre for Integrated Mountain Development (ICIMOD), Kathmandu, 44700, Nepal

[5]National Center for Atmospheric Research (NCAR) Boulder, 80301, USA

[6]Physical Research Laboratory (PRL), Ahmadabad, 380009, India

Key words: Kathmandu, Himalayas, Air Pollution, Ozone, CO, Long Range Transport

*Correspondence to*: Manish Naja (manish@aries.res.in)

**Highlights of the study:**

- **A regional picture of the SusKat field campaign based on synergistic analysis of trace gas observations from Kathmandu Valley and India.**

- **Regional transport also contribute to the springtime ozone enhancement in the Kathmandu Valley.**

- **Biomass burning in NW IGP led to simultaneous increase in $O_3$ and CO levels in the Kathmandu Valley and northern Indian sites.**

**Abstract**

Air pollution resulting from rapid urbanization and associated human activities in the Kathmandu Valley of Nepal has been leading to serious public health concerns over the past two decades. These concerns led to a multinational field campaign SusKat-ABC (Sustainable atmosphere for the Kathmandu Valley-Atmospheric Brown Clouds) that measured different trace gases, aerosols and meteorological parameters in the Kathmandu Valley and surrounding regions during December 2012 to June 2013 to understand local to regional scale processes influencing air quality of the Kathmandu Valley. This study provides information about the regional distribution of ozone and some precursor gases using simultaneous in situ measurements from a SusKat-ABC supersite at Bode, Nepal and two Indian sites: a high-altitude site Nainital located in the central Himalayan region and a low altitude site Pantnagar located in the Indo-Gangetic Plain (IGP). The diurnal variations at Bode showed a daytime buildup in $O_3$ while CO shows morning and evening peaks. Similar variations (with lower levels) were also observed at Pantnagar but not at Nainital. Several events of hourly ozone levels exceeding 80 ppbv were also observed at Bode. The CO levels showed a decrease from their peak level of about 2000 ppbv in January to about 680 ppbv in June at Bode. The hourly mean ozone and CO levels showed a strong negative correlation during winter ($r^2$=0.82 in January and $r^2$=0.71 in February), but this negative correlation gradually becomes weaker, with the lowest value in May ($r^2$=0.12). The background $O_3$ and CO mixing ratios at Bode were estimated to be about 14 ppbv and 325 ppbv, respectively. The rate of change of ozone at Bode showed a more rapid increase (~17 ppbv/hour) during morning than the decrease in the evening (5-6 ppbv/hour), suggesting prevalence of a semi urban kind of environment at Bode. The lower CO levels during spring suggests that regional transport also contributes appreciably to springtime ozone enhancement in the Kathmandu Valley on top of the local in situ ozone production. We show that regional pollution resulting from agricultural crop residue burning in north-western IGP led to simultaneous increases in $O_3$ and CO levels at Bode and Nainital during first week of May 2013. Biomass burning induced increase in ozone and related gases was also confirmed by a global model and balloon borne observations over Nainital. A comparison of surface ozone variations and composition of light non-methane hydrocarbons among different sites indicated the differences in emission sources of the Kathmandu Valley and the IGP. These results highlight that it is important to consider regional sources in air quality management of the Kathmandu Valley.

## 1. Introduction

The Himalayan region is among the least studied regions in the world despite its known importance in influencing the livelihood of about a billion people and agricultural systems. The Himalayas are spread over a large region from Afghanistan, Pakistan, India, Nepal, Bangladesh, Bhutan, China, and Myanmar, and provide fresh water to about a billion people living in this region. However, the growing economies, industrialization and increasing population in the region are polluting this pristine environment and perturbing the regional environment, climate and ecosystems. The urban centers in the mountain regions often face severe air pollution problems since the mountains act as a barrier to horizontal ventilation of the pollutants and local mountain valley winds govern the diurnal variations in air pollutants. These processes have been well studied over other parts of the world, such as Mexico City (de Foy et al., 2006; Molina et al., 2007 etc.), Po Valley (Martilli et al., 2002) and Santiago de Chile (Schmitz, 2005; Rappengluck et al., 2005). The Kathmandu Valley, located in the central Himalayas is an ideal natural laboratory to study such processes. However, only a few surface measurements of ozone and related trace species have been reported so far from this region (Pudasainee et al., 2006; Panday and Prinn, 2009; Christofanelli et al., 2010; Putero et al., 2015; Mahata et al., 2017).

The valley has experienced an unprecedented growth as the population increased nearly fourfold from about 0.75 million to about 3 million over the last 25 years. The total vehicle fleet in the Bagmati Zone, where the Kathmandu Valley is situated, increased by about 22 times from about 34,600 in 1989-90 to about 755,000 vehicles in 2013-14 (DoTM, 2015; http://www.dotm.gov.np/en). Consequently, the total fossil fuel usage in the valley is about 50% of all of Nepal. The shares of coal, petrol, diesel kerosene and liquefied petroleum gas (LPG) usage

in the Kathmandu Valley ranges between 35% and 66% when compared with their respective usage in all of Nepal (Pradhan et al., 2012). These unprecedented growths can have serious implications for the air quality and its impacts in Kathmandu Valley, such as higher occurrences of respiratory problems, skin and eye irritation have already been observed among the people

living in the Kathmandu Valley than in other areas (Pradhan et al., 2012). In the past, elevated levels of $O_3$, CO, $NO_x$ and VOCs have been reported over this region during winter and pre-monsoon seasons (Pudaisanee et al., 2006; Pandey and Prinn, 2009).

In order to advance our understanding of air quality in the Kathmandu Valley and to understand

how the magnitude and variability of air pollution in Kathmandu Valley compares with other sites in the region, the Sustainable Atmosphere for the Kathmandu Valley- Atmospheric Brown Clouds (SusKat-ABC) international air pollution measurement campaign was carried out in Nepal during December 2012-June 2013, with an initial intensive measurement period of two months from December 2012 to February 2013 (Rupakheti et al., 2017). Eighteen international research groups

participated and various instruments for the extensive measurements of aerosols, trace gases and meteorological parameters were installed. The campaign covered a total of 23 sites of various measurement capabilities in the region with a supersite at Bode, 5 satellite sites in and on the Kathmandu Valley's rim, 5 regional sites (Lumbini, Pokhara, Jomsom, Dhunche and Pyramid) and other collaborating sites in India and China, including Nainital and Pantnagar in India.

Measurements of short-lived climate-forcing pollutants (SLCP), i.e., ozone and black carbon at Paknajol near the city center of Kathmandu during the SusKat-ABC campaign are reported in Putero et al. (2015). However, that study lacked the collocated measurements of $O_3$ precursors. Sarkar et al., (2016) presented the measurements of non-methane volatile organic compounds

(NMVOCs) at one-second resolution using the Proton Transfer Reaction-Time of Flight-Mass Spectrometry (PTR-TOF-MS) and study on two greenhouse gases are described by Mahata et al., (2017) at Bode during the campaign. These studies provided important information about atmospheric composition in the Kathmandu Valley during the SusKat-ABC period; however, a regional picture of the variability in ozone and related gases has not been presented so far.

In light of the above conditions, this study aims to provide first information about the regional distribution of ozone and related gases during the SusKat-ABC by synergistically analyzing simultaneous in situ measurements of surface ozone and CO at Bode from January to June 2013 with those from two Indian sites, namely Nainital (a high altitude site in the central Himalayas) and Pantnagar (a low altitude site in the Indo-Gangetic Plain (IGP)). Additional observations at the two Indian sites are used to understand the similarities and differences between the air quality of Kathmandu Valley and the Indian sites, and identify any regional emission source common to these sites. The previous measurements of ozone in the Kathmandu Valley were only performed near the city centers; however, Bode is on the eastern side of the valley and is generally downwind of the major urban centers of Kathmandu Valley (Kathmandu Metropolitan City and Lalitpur Sub-metropolitan City). This site also receives regional air masses from west and south especially during afternoons with stronger wind speeds. Therefore, this site can serve as a better representative to suggest the background levels of $O_3$ and CO in the Kathmandu Valley.

## 2. Experimental details

### 2.1. Observation sites

The Kathmandu Valley is an oval-shaped urban basin located in the central Himalayan foothills between the IGP and the Tibetan Plateau (Figure 1). The valley is surrounded by mountain peaks with altitude ranging from 2000 to 2800 m above mean sea level (amsl) and five mountain passes (Nagdhunga, Bhimdhunga, Mudku Bhanjhyang in the West, Sanga and Nagarkot in the East) with altitude ranging from 1500 to 1550 m amsl, and the outlet of the Bagmati River in the southwest corner of the Valley. The flat base area of the Kathmandu Valley is about 340 $km^2$ with a mean elevation of about 1300 m amsl. There is no river inlet into the Kathmandu Valley and only one narrow river outlet (Baghmati river) in the southwestern side. The spatial extent of the valley is about 25 km in East-West and around 20 km in the North-South direction. The measurement sites located in Kathmandu Valley during the SusKat field campaign are depicted in Figure 1b. In this study observations of $O_3$, CO and meteorological parameters made at Bode (27.68º N, 85.39º E, 1344 m amsl) and at two sites in India viz., ARIES, Nainital; a high altitude site located on a mountain top (29.36º N, 79.45º E, 1958 m amsl) and Pantnagar; located in the Himalayan foothills in IGP (29.0º N, 79.5º E, 231 m amsl) are discussed.

Two observations sites in India, i.e., Nainital and Pantnagar represent cleaner Himalayan and polluted IGP environments, respectively. Surface ozone levels at Nainital are found to be driven mainly by transport of anthropogenic emissions from the IGP region (Kumar et al., 2010; 2011), while those at Pantnagar are mostly controlled by local emissions (Ojha et al., 2012). In the previous studies, the pollutants level in the Kathmandu valley were reported to be primarily

influenced by the local emissions and the unique meteorology in the region (Panday et al., 2009; Putero et al., 2015).

### 2.2. Ozone and CO measurements

Surface ozone measurements  are conducted using two types of analyzers, i.e., Teledyne M400E (at Bode and Pantnagar), and Thermo Model-49i (at Nainital). The observation principle of both the instruments is based on the commonly used technique of attenuation of UV radiation (~254 nm) by ozone molecules. These instruments are regularly subjected to zero and span tests using an internal ozone generator and ozone observations from both the instruments are also inter-compared

by running them side by side and using a common inlet. Further details of such inter-comparisons are reported in Sarangi et al. (2014).

CO measurements are also conducted using two types of analyzers i.e., Horiba APMA-370 (at Bode and Pantnagar) and Thermo 48i (at Nainital). CO instrument, at Bode, was deployed for the

first time in field after factory calibration from the manufacture. Nevertheless, both CO instruments were inter-compared using a common inlet prior to the campaign and correlation coefficient between CO mixing ratios measured by the instruments is estimated to be ~0.9 with a slope of 1.09. CO instruments are based on commonly used method of infrared (IR) absorption by the CO molecules at 4.6 $\mu$m. Regular zero check and span check for CO instruments are performed

using a primary calibration mixture  from Linde UK (1150 ppbv; Sarangi et al., 2016) and secondary gas from Chemtron Science Laboratories (1790 ppbv). Multipoint calibrations (ultra-pure gases) are also carried out in different observational ranges using a zero air generator (Thermo model 1160) and a dynamic gas calibrator (Thermo model 146i) (Sarangi et al., 2014). The

meteorological measurements at Bode are performed using an automatic weather station (Campbell Scientific, UK).

Both the $O_3$ and CO instruments were installed on the fourth floor of a building in Bode facing eastern side of the Kathmandu Valley (Figure 2; refer Sarkar et al., 2016 for site description). The sampling inlet for these instruments was placed at the top of the building and Teflon (TFE) tubes were used for the air intake. The $O_3$ and CO instruments at Nainital and Pantnagar were placed in atmospheric science building at Manora Peak [refer Sarangi et al., 2014 for site description] and at the College of Basic Sciences and Humanities (CBSH), G. B. Pant University of Agriculture and Technology (GBPUAT), at Pantnagar (refer Ojha et al., 2012 for site description). Observations at Bode are in Nepalese Standard Time (NST), which is 5:45 hours ahead of GMT, and observations at Nainital and Pantnagar are in Indian Standard Time that is 5:30 hour ahead of GMT.

**2.3. Air sampling and analysis for hydrocarbons**

A total of 16 air samples are collected at Bode from 30 December 2012 to 14 January, 2013 with the frequency of one sample per day. These air samples are collected at 1400 hour (two samples at 1200 hour) when the boundary layer is fully evolved and the air is well mixed. Air samples are collected at a pressure of 1.5 bar and analyzed for ethane, ethane, propane, i-butane, n-butane, acetylene and i-pentane using a gas chromatograph (HP 5890 II) equipped with a flame ionization detector and a PLOT column of $KCl/Al_2O_3$. Helium is employed as the carrier gas and $H_2$ & zero-air are used for flame. Air samples are also analyzed for $CH_4$ and CO using another GC (Varian Vista, 6000, USA) and employing a molecular sieve 13x, packed column (4 m). CO is measured

by converting in to $CH_4$ using a Ni catalyst heated to about 325ºC. Standard mixture from Intergas (International Gases & Chemicals), UK traceable to National Physical Laboratory (NPL), UK, is employed for calibration of NMHCs. Gases from NIST, USA and Linde, UK are used for calibration of $CH_4$ and CO. More details on sample pre-concentration and calibration can be seen in Lal et al., (2008), Mallik et al., (2014) and Sarangi et al., (2016).

**2.4. Satellite data emissions, and back-air trajectory**

In this study, Ozone Monitoring Instrument (OMI) level-3 daily tropospheric column amount $NO_2$ data product OMNO2d (cloud screened at 30%) at 0.25º x 0.25º resolution is used to generate spatial maps during biomass burning period. This product is based on the radiance measurements made by OMI instrument in visible (VIS, 405-465nm) channels. A detailed description of measurement principle can be found in Bucsela et al., (2013). The fire locations during the spring time biomass burning are used from monthly MODIS collection 6, Level 2 (combined Aqua and Terra) global monthly fire product mcd14ml at 1 km resolution [Giglio,2010]. For this study fire locations with high detection confidence (>80%) are used and detailed detection principle can be found in Giglio et al. (2003) and Justice et al., (2006). NASA's Atmospheric Infrared Sounder (AIRS) instrument suite measures atmospheric water vapor and temperature profiles on a global scale. At present the operational instruments in this suite consists of a hyperspectral infrared instrument (AIRS) and a multichannel microwave instrument Advanced Microwave Sounding Unit (AMSU-A). Apart from meteorological datasets these instruments also retrieves the vertical profiles of some of the trace gases such as ozone and CO and detailed description of retrieval algorithms are discussed in Susskind et al., (2003, 2006). In this study the vertical profiles of ozone, CO and relative humidity from AIRS+AMSU joint data product (AIRX3STD v006) at 1 degree

spatial resolution are used to understand stratosphere-troposphere transport phenomenon during Jan-May 2013.

The biomass burning emissions of CO were estimated using Global Fire Emissions Database
(GFED) version 4 datasets. This data uses satellite information of fire hotspots, vegetation productivity to calculate gridded fire emissions. The data also included fractional contribution of different types of vegetation to fire emissions. The version 4 dataset has spatial resolution of 0.25 degrees, and detailed description about the dataset and data access could be found from http://www.globalfiredata.org/data.html. The Modern-Era Retrospective analysis for Research and
Applications, Version 2 (MERRA-2) is the reanalysis dataset produced by Global Modeling and Assimilation Office (GMAO) of NASA. In this study ertel potential vorticity data is used from this reanalysis and more details can be found in Gelaro et al., (2017).

To understand the history of air masses arriving at Bode, Hybrid Single Particle Lagrangian
Integrated Trajectory (HYSPLIT) model (Draxler and Hess, 1999) is used. The high resolution $(0.5^{o} \times 0.5^{o})$ meteorological fields from Global Data Assimilation System (GDAS) 0.5 degree is used to generate a 5-day backward trajectories over Bode. These trajectories are initiated as an ensemble of 9 points separated by 0.25 degrees around Bode at 1km above ground level for 4 different months.

### 2.5. WRF-Chem Simulations

WRF-Chem version 3.5.1 with two-nested domains was used for this simulation. The coarse domain that encompasses the area between 16-43° N and 68-107° E, was use with 15-km grid

spacing, the nested domain that covers the central part of Nepal and the Kathmandu Valley with 3-km grid spacing,  and 35 vertically-stretched layers from the ground up to 50 hPa (Mues et al., 2017). The physics options used for this study include the Lin microphysics scheme (Lin et al., 1983), the Grell cumulus parameterization (Grell and Dévényi, 2002), the Rapid Radiative

Transfer Model (Iacono et al., 2008) for longwave and Goddard shortwave scheme (Chou and Suarez, 1994), the Yonsei University boundary-layer parameterization (Hong et al., 2006), and the MM5 scheme for the surface layer (Jimenez et al., 2012).The initial and boundary condition for meteorological fields are using the ERA-Interim data. Anthropogenic emissions were obtained from the HTAP V2 inventory (http://edgar.jrc.ec.europa.eu/htap_v2). Emission maps for CO and

NOx are shown in supplementary material (Figure S1 and S2). The RADM2-SORGAM chemical mechanism is used to represent the gas-phase and aerosol chemistry. The photolysis rates were computed using the Fast Tropospheric Ultraviolet and Visible (FTUV) Radiation Model (Tie et al., 2003, Li et al., 2005). The dry deposition was calculated following Wesely (1989) resistance method.  Biogenic emissions were computed on-line using the Model of Emissions of Gases and

Aerosols from Nature (MEGAN) model (Guenther et al., 2006). The biomass burning emissions are based on Fire INventory from NCAR (FINN) (Wiedinmyer et al., 2011).

## 3    Results and Discussion

### 3.1 General Meteorology

The Kathmandu Valley is located in the central Himalayan region due north of the IGP and south of the Tibetan Plateau. The valley is influenced by the South-Asian monsoon and in general receives most of its precipitation during summer (June-September). The remaining seasons are relatively dry with spring or pre-monsoon season (March-May) being the hottest (Panday and

Prinn, 2009). Figure 3 shows the diurnal variations in temperature, relative humidity (RH), solar radiation, wind speed and wind direction at Bode during January and April 2013, which are chosen as representative months for winter (Jan-Feb) and spring (Mar-May), respectively. The average daily RH during January (67%) was higher than in April (62%), with the diurnal maximum during early morning hours. During January, high RH in early morning hours (92% during 5-7 AM) was associated with the foggy conditions during morning hours.

The average temperature and solar radiation were higher in April (20.7$^\circ$ C and ~800 W/m$^2$ ) than January (11.4$^\circ$ C and ~600 W/m$^2$). The diurnal variations in temperature showed highest values during late afternoons which was few hours after the peak in diurnal solar radiation. Solar radiation strongly influence the diurnal variations in temperature and other meteorological parameters by surface heating causing thermals to rise. The wind speeds were the slowest (<1 m/s) during the night and early morning hours, primarily easterlies, and the highest wind speeds were observed during mid-late afternoon, primarily westerlies (4-6 m/s). These wind patterns were similar in both the months with April having slightly longer duration of daytime westerly flow. These wind flows, together with boundary layer dynamics are responsible for the dispersion/accumulation of pollutants during the course of a day.

The other sites in the Indian region viz., Pantnagar and Nainital also show high values of solar radiation and temperature during spring months and lowest during winter season (Ojha et al., 2012; Sarangi et al., 2014; Naja et al., 2016). The Nainital site which is at a remote mountain top experiences moderate northwesterly winds (~2-3 m/s) during most of the year with prevalence of southeasterly winds during the summer monsoon period. The Pantnagar site is situated in the

vicinity of Himalayan foothills in the IGP and the similar seasonal changes in temperature, RH and solar radiation are observed (Ojha et al., 2012). In the summer monsoon season, the lowest levels of $O_3$ and CO are observed due to arrival of cleaner marine air masses. During winter months, slow winds, poor ventilation and the lowest boundary layer heights are observed, and during the first week of January several cases of widespread fog are also observed.

### 3.2 Back-air Trajectories

To understand the wind patterns over Bode, the HYSPLIT model is used to generate back-air trajectories. Fig. 4 shows the monthly averaged nine-point trajectories (in different colors) for January, March, May and June around Bode. During winter and early spring months of January and March, a strong westerly flow is observed with air masses passing through IGP prior to entering Nepal region. During late spring (May) these westerly air masses slow down and a slight change in directions are observed as some of the air masses passes through the central part of India. The air-masses were observed to be mostly arriving from higher altitude during winter than during spring season (not shown in Fig. 4). Similar behavior of air masses is also observed over Nainital (Kumar et al., 2010). However, during June, the monsoon circulations takes over and reversal in wind directions are observed (Fig. 4d).The above features for Bode are similar to those seen over Nainital and Pantnagar. The back air trajectories analysis for Nainital and Pantnagar shows that long-range transport (westerly wind) is a vital factor during winter. However, air masses mostly circulate over the continental northern Indian region at low altitudes during spring and autumn seasons when local pollution plays an important role (Kumar et al., 2010; Ojha et al., 2012; Naja et al., 2016).

**3.3 Variations in ozone at Bode**

The monthly (January to June) average diurnal variations in ozone at Bode are shown in Figure 5. The diurnal variations in ozone show higher levels during daytime. This daytime build-up in ozone is consistently observed throughout the observation period, with relatively lesser buildup during June due to prevailing cloudy/rainy conditions. Additionally, there are only a few days of observations in June, when the campaign ended. The daytime increment in surface ozone is a typical feature of polluted sites and can be associated with daytime photochemical production of ozone from its precursors in the presence of sunlight (e.g., Kleinman et al., 1994). Model simulated normalized average diurnal variations also show a clear daytime buildup in ozone during February and May (Figure S3 in supplementary material).

The very low levels of ozone (during winter months) were observed during night-time which can be attributed to titration of $O_3$ by NO. The boundary layer measurements made using a ceilometer during the campaign suggests lower heights during night time (Figure 6) with very low ventilation coefficients during winter (Mues et al., 2017). The sampling inlet for the gaseous measurements was on the rooftop about 20m from the ground level, the possibility of loss of ozone due to surface deposition in highly stratified nighttime boundary layer should be much less. Unfortunately, observations of NO and NO2 are not available during the campaign but some information about the levels of these species can be obtained from the previous studies. Pudasainee et al. (2006) measured the NO, $NO_2$ during winter of 2003-2004. Yu et al (2009) also measured the NO, $NO_2$ and HONO during the similar period and showed two peaks at morning (07:00-08:00) and evening (19:00-20:00) in NO with maximum levels reaching as high as 60 ppb. Although the results are almost a decade old and the difference in $O_3$ magnitude is also observed (Figure S4 in

supplementary material) but we expect the increase in NO levels over the valley because satellite retrieved tropospheric column $NO_2$ show an increase over this region (Figure S4).

Just around sunrise, a dip in ozone levels was observed. A similar dip in ozone was observed at an urban site in India (Lal et al., 2000) which is suggested to be due to its reaction with NO and $NO_2$. Since, observations of NO, $NO_2$, $NO_3$ and $N_2O_5$ are not made during the campaign, we have employed model results to explain this feature. Figure 7 show model simulated average diurnal variations in NO, $NO_2$, $NO_3$ and $N_2O_5$ during February and May 2013. NO mixing ratios are close to zero during the nighttime because it rapidly reacts with $O_3$ to form $NO_2$, which also explains higher $NO_2$ levels during nighttime. $NO_3$ and $N_2O_5$ also show higher levels during nighttime because of the reactions of $NO_2$ with $O_3$, of $NO_2$ with $NO_3$, respectively. The sharp morning increase in NO mixing ratios correlates strongly with the sharp decrease in $NO_3$ and $N_2O_5$ mixing ratios especially during February indicating that photodissociation of $NO_3$ ($\lambda$ <670 nm) and $N_2O_5$ (280< $\lambda$ <380nm) releases NO back to the atmosphere. It is evident that NO3 and N2O5 peaks prior to NO and NO2 and later they are removed.

At Bode, spring time higher levels of ozone with a broader peak was observed when compared with winter months, this can be attributed to the increase in incoming solar radiation which in turn increases the photochemical production of ozone. Role of air-masses (local Vs regional contribution) will be discussed later in subsequent sections where differences in winter and spring variations will be clearer. The nighttime ozone mixing ratios during the campaign are the lowest during the winter season. During this period, lower boundary layer height was also observed (Figure 6) (Mues et al., 2017). The ventilation coefficient is a measure of transport and/or mixing

of pollutants in the boundary layer. Due to very low wind speeds (<1 m/s) and shallow boundary layer height during night, low ventilation coefficients (<100 $m^2$/s) were observed during this season (Figure S5).

**3.4 Variations in CO and Hydrocarbons at Bode**

The monthly average diurnal variations in CO showed two peaks, one during morning and the other in the evening hours (Figure 5). The monthly mean values of $O_3$ and CO with standard deviation, maximum and minimum are given in Table 1. Averages of both these gases in four time periods are also given in Table 2. Diurnal variation in CO with two peaks is a typical pattern over a polluted site and such variations have been reported at different South Asian urban sites e.g., Ahmedabad (Lal et al., 2000), Kanpur (Gaur et al., 2014), Pune (Beig et al., 2007), Santiago de Chile (Rappengluck et al., 2005), and Chicago (Pun et al., 2003) etc. The major sources for CO in the valley are vehicular emissions, brick kiln emissions, domestic burning of biofuels for cooking and heating, and garbage burning etc. Out of these, vehicular emissions, cooking and heating occurs largely during two times a day, morning and evening but with some time differences. Additionally, the nighttime and early morning hours are characterized by very slow wind speeds, and a shallower boundary layer along with poor mixing.

It is to be noted that CO levels during morning peaks are greater than those during evening. Such higher peaks during morning times than those during evening times have also been observed by Panday and Prinn, (2009) and have been explained on the basis of overnight accumulation of CO emissions. During daytime the emissions are countered by the boundary layer evolution and dynamic processes such as flushing of CO and other pollutants by westerly winds (Figures 1 and 3) blowing throughout the afternoon across the valley through the eastern passes. The chemical

loss of CO via reaction with OH radical could also contribute to lower daytime CO levels. However, OH measurements are not made during the campaign and it is not possible to confirm this aspect.

The mixing layer starts evolving after sunrise and reaches its peak median values of about 900-1200m during winter and spring seasons (Mues et al., 2017). The ventilation coefficient estimated (with 15 m wind data) during daytime suggested vertical mixing could occur from mid-afternoon till early evening hours when consistently high VC (> 1000 $m^2$/s) was observed (Figure S3). During this period the daily maximum wind speeds (4-5 m/s at 15:00-17:00) were also observed

which were mostly westerly and further decreases the CO levels with daily minimum levels (148-218 ppbv) observed during this period. Due to relatively higher wind speeds and associated advection before evening hours, the CO mixing ratios at 19:00-20:00 are about 30-50% lower than morning hours.

In addition to CO, observations of VOCs made at Bode (Sarkar et al., 2016) also showed similar diurnal variations with two peaks and having their levels up to about 15 ppbv. Similar to CO, many VOCs (e.g. Acetonitrile, Benzene, Furan, etc.) showed higher levels during morning, when compared to the evening peaks. We have also collected one air sample almost every day during 30 December 2012 – 14 January 2013 and analyzed them for light non-methane hydrocarbons ($C_2$-

$C_5$) and for CH4-CO. Average values along with standard deviation, minimum, maximum and number of samples are given in Table 3. Average methane levels for the measurement period at Bode is 2.55±.12 ppmv which are much higher than the global average and about 28% and 27% higher than measured at a Northern Hemisphere background site at Mauna Loa (1.84 ppmv), and

at a remote site Mt. Waliguan, China (1.87ppmv), respectively (www.esrl.noaa.gov). The average values of other hydrocarbons varied from about 1 ppbv to about 4.4 ppbv. The maximum value is observed to be of propane (15.48 ppbv) and acetylene (14.35 ppbv). These highest mixing ratios are observed on 7 January 2013.

The mixing ratios of eight light non-methane hydrocarbons at bode are relatively higher (ppbv) than those observed at Nainital (0.8-2.2 ppbv) and comparable with those at Pantnagar/Haldwani (0.8-3.7 ppbv and 1.6-4.2 ppbv respectively). Additionally, unlike Bode, none of these sites in India showed NMHC mixing ratios exceeding 10 ppbv. Methane mixing ratio is also higher at

Bode (2.55 ppmv) than those observed at Nainital (1.89 ppmv). Figure 8 shows a comparison of contribution of eight light NMHCs at Bode, Nainital, Pantnagar (including another town Haldwani) (Sarangi et al., 2016) and Kanpur (Lal et al., 2008). Bode data are for December-January months, while data from rest of three sites are in December month. Composition at Bode shows difference with those at Indian sites. Propane (20%) and n-butane (13.5%) show greater

contribution, while contribution of i-pentane (4.2%) is significantly lower at Bode when compared with India sites. Greater contribution of propane and n-butane indicates for some leakages of liquefied petroleum gases (LPG) in the Kathmandu Valley.

**3.5 Correlation between ozone and CO**

It has been discussed in the previous section that $O_3$ and CO show some contrasting diurnal variations. Here we discuss about the correlation between $O_3$ and CO during different months (Figure 9). The highest negative correlation is seen in winter period ($r^2$=0.82 in January and $r^2$=0.71 in February) and this negative correlation reduces gradually with the lowest value in May

($r^2$=0.12). The reduction in the tendency of the negative correlation from January to May could be due to changes in emission patterns and the boundary layer mixing. This is discussed further in the subsequent paragraphs. Hourly average CO levels show a systematic decrease from ~2300 ppbv in January to about 680 ppbv in June, whereas ozone shows a tendency of increase. The daytime ozone levels during spring season are slightly higher (~62 ppbv) when compared to winter (~54 ppbv). This spring time increase in ozone levels is also reported by several other studies in northern part of the Indian subcontinent (Kumar et al., 2010; Ojha et al., 2012; Kumar et al., 2013; Gaur et al., 2014). The higher values of CO during winter season can be attributed to an increase in its emissions (domestic and garbage burning to keep warm in winter season) and their inefficient dilution due to poor mixing and shallower boundary layer (Mues et al., 2017). However, during spring season, greater and relatively well mixed daytime boundary layer leads to show lower CO levels. As mentioned previously, role of OH chemistry could be an additional contributor in lower CO levels in spring. Model simulated OH levels are found be higher in May, when compared with February. This would suggest great chemical loss of CO in spring (May).

Figure 10 shows daily variations in $O_3$ and CO during four different times *i.e.,* 0300-0500 hours, 0730-0830 hours, 1300-1500 hours, and 2200-2300 hours. It is considered that 0300-0500 hours and 2200-2300 hours would provide information for the periods when photochemical production of ozone is absent, while 1300-1500 hours can be used to understand the behavior during the periods of high photochemical activity and fully evolved daytime mixed layer. Variations during 0730-0830 hours will provide the information during morning period. The stable nocturnal boundary layer starts evolving during morning hours and air mass close to surface begins to mix with air at higher heights. In general, CO levels (blue line) show a decrease from January to June

during 0300-0500 hours, 0730-0830 hours, and 2200-2300 hours, while they do not show significant changes during 1300-1500 hours. In-contrast, ozone levels (red lines) are increasing from January to May/June during all four time periods. The highest noontime ozone level is observed to be about 80 ppbv during January to March that increases to about 102 ppbv during April-May. The noontime ozone level comes down to about 46 ppbv in June, which is mainly due to beginning of the monsoon season that is characterized by the arrival of cleaner air from the oceanic regions of the Bay of Bengal, the Arabian Sea and the Indian Ocean.

The increase in ozone from January to May is rather more during nighttime or early morning hours, when photochemical production of ozone is absent. This suggests an enhancement in the background ozone levels. Hence, probably, average ozone value ($13.1 \pm 1.2$ ppbv) during early morning period of 0300-0500 hours would represent background ozone levels for this region. However, the emissions are directly influencing the CO levels. Hence, the estimated CO mixing ratio ($325.4 \pm 98.3$ ppbv) during noontime (1300-1500 hours) could be considered as background levels for the Bode region. Since the noontime boundary layer at Bode is assumed to be well mixed and the fast westerly flows across the valley reduces the direct sampling of air-masses at Bode from its immediate emission sources scattered in the valley.

Figure 10 also shows correlation between $O_3$ and CO for different time periods. Weak negative correlation is seen during early morning (0300-0500 and 0730-0830 hours) or night hours (2200-2300 hours). During nighttime and early morning hours, lowest boundary layer height (150-200m) was observed. In an urban or semi-urban environment of Kathmandu valley, where $NO_x$ levels are not lower (Pudasainee et al., 2006) ozone titration takes place throughout nighttime. Figure 10 also

indicate the increase in overnight CO levels which peaked during morning hours. These contrasting variations tends to show the negative correlations. However, a slight positive correlation is observed during noon period (1300-1500 hours) which is similar to what is generally, observed at high altitude sites (Kaji et al., 1998; Tsutsumi and Matsueda, 2000; Naja et al., 2003; Sarangi et al., 2014) and cleaner sites (e.g. Island sites, Pochanart et al., 1999).

### 3.6 Regional distribution of $O_3$ and CO during SusKat

Apart from $O_3$ and CO observations at Bode, simultaneous observations of these two gases were also made at the central Himalayan site in India (Nainital) and a site in Himalayan foothills in the IGP region (Pantnagar) and are discussed in this section. Pantnagar and Nainital, despite being different in altitude, the wind patterns over this region are mostly northerly or northwesterly during winter (Kumar et al., 2010; Ojha et al., 2012; Sarangi et al., 2014). Further, both the sides receive polluted air massed from the IGP during spring season. Whereas the cleaner marine air masses arrive at these sites during the summer-monsoon season. Average diurnal patterns in $O_3$ and CO mixing ratios are similar at Bode and Pantnagar with twin peaks in CO and daytime high levels of $O_3$. However, different variations (Figure 11) are observed at Nainital (green line), which being a remote high altitude site does not show any daytime photochemical buildup or nighttime loss in ozone. Further, the daytime ozone levels at Bode are higher than those at Pantnagar during winter season, while these are comparable during spring season. Additionally, CO levels are also higher at Bode than those at Pantnagar during winter (Figure 11). A comparison of surface ozone measurements at Kanpur (India) showed a relative better agreement with Pantnagar (India) while measurement at Paknajol (Nepal) showed better agreement with Bode (Nepal) during both seasons

(winter and spring) (Figure 11) indicating the differences in emission sources of Kathmandu Valley and the IGP.

The changes in ozone increase/decrease rates (ppbv/hour) are analyzed for all these five sites. Generally, the ozone increase/decrease rates are nearly symmetric during morning and evening at an urban site. However, it is asymmetric with slower changes occurring during afternoon/evening time at a rural or semi-urban sites (Naja and Lal., 2002). Ozone production is strongly dependent on amount of precursor gases and available sunlight. On the contrary, evening time ozone loss depends mainly upon its titration with NO, apart from surface deposition. This rate of change of ozone during morning and evening hours has been used as an indicator of chemical environment (rural or urban) over a site (e.g., Naja and Lal., 2002). Below, we discuss calculated ozone increase/decrease rates.

Figure 12 shows that the wintertime rate of ozone increase in morning hours is much higher at Bode (about 17 ppbv/hour), when compared to Pantnagar (about 9 ppbv/hour). This suggests a rapid ozone buildup at Bode than at Pantnagar. In contrast, the ozone decrease rate is lower at Bode (5-6 ppbv/hour) when compared to the decrease rate at Pantnagar (about 14 ppbv/hour) during spring. This suggests rather slower ozone loss at Bode via NO titration, indicating somewhat lesser polluted kind of environment in Bode during spring. However, this does not necessarily mean that $NO_x$ emissions are lower in the Kathmandu Valley. NO levels are reported to be as high as 60 ppbv (Yu et al. 2009). Another process driven by diurnal variations in winds could lead to slower evening ozone loss rates. Faster afternoon westerly winds flush the pollutants out of the valley every day, leaving less $NO_x$ to titrate ozone during evening hours. In contrast,

slower winds at night allow overnight accumulation of precursor gases in nocturnal boundary layer of Kathmandu Valley that in turn can potentially contribute to next morning ozone build-up. Similar to the diurnal variations in average ozone, diurnal pattern in ozone change rates are similar at Pantnagar and Kanpur.

Back-air trajectory assisted analysis of ozone observations at Nainital in the central Himalayas show that the major role of regional/local pollution is in spring when regionally polluted ozone levels are estimated to be 47.1± 16.7 ppbv (Kumar et al., 2010). During spring, net ozone production over the northern Indian Subcontinent is estimated to be 3.2 ppbv/day in regionally

polluted air masses in spring but no clear build-up is seen at other times of year. While the role of long-range transport is shown to be important in winter with contribution of about 8-11 ppbv of ozone.

Shorter duration of solar heating during winter leads to weaker dynamical processes including

convective mixing of pollutants, which in turn confines the pollutants near to the surface. Additionally, the Kathmandu Valley is isolated inside the Himalayas and the only way for pollutants to reach here is either via upslope flow of polluted air masses through the mountain valleys or arrival of polluted regional air masses from the air aloft. Thus, the wintertime $O_3$ and CO mixing ratios at Bode to be unlikely influenced by IGP pollution. Further similar trapping of

pollutants during winter season are also reported by previous studies done over this region (Panday and Prinn, 2009). However, intense heating and stronger convective mixing could induce the IGP outflow to influence this valley region during spring season. Spring time ozone enhancement, due

to IGP outflow, in the tropical marine region surrounding India has already been observed (Lal et al., 2013).

### 3.7 Influences of springtime northern Indian biomass burning

Every year northern Indian biomass burning emits large quantities of trace gases and aerosols and significantly affect the regional distribution of several trace species (Kumar et al., 2011; Sinha et al., 2014; Putero et al., 2014; Bhardwaj et al., 2016; Kumar et al., 2016). These studies showed the enhancement in $O_3$ and CO levels due to crop residue and forest fire burning in the IGP during pre- and post-monsoon seasons. To investigate the influences of biomass burning on the surface ozone and CO mixing ratios over Bode, a time series of high confidence (confidence > 80%) fire counts over the northern Indian subcontinent (25º-35ºN, 70º-95ºE) were analyzed. Based on MODIS fire counts, the fire activity period is chosen to be April 3 to May 31, 2013.

High fire activity period (HFAP) is defined when 3-day running mean of fire counts exceeds the median fire counts during the fire activity period (Kumar et al., 2011). The low fire activity period (LFAP) is defined as March 1-31 when very low fire counts are observed over northern Indian subcontinent. The changes in surface ozone and CO mixing ratios during these two periods (HFAP and LFAP) are shown in figure 10 (right panel: center and bottom). The average fire emissions over the Kathmandu region (27º-28ºN, 85º-86ºE) using GFED v4.0 emission inventory are also calculated and shown in Figure 13 (top right). Since the HFAP is almost 4 weeks long, two different peaks in ozone and CO mixing ratios during April and May are selected (see highlighted peaks in Figure 13). During both the periods a prior-increase in fire counts were observed which is followed by higher levels of ozone and CO mixing ratios. Another reason to separately study

these cases, is fire locations during these two periods. High CO and $O_3$ case in April are shown to be induced by the fires those are mostly located in Nepal region (dark red), while fire activity was very low in the northern Part of the IGP in April (Figure S6). Whereas, during May event (orange), high increase in fire counts (~ 2.5 fold) is observed over northern Indian subcontinent during the HFAP (Figure S6) and very low fire was seen in Nepal region. Therefore, these two cases could be studied separately to assess impacts of biomass burning over Bode and other sites. The changes during HFAP, LFAP and above two event cases in April and May are shown in Table 4.

During the first half of April daily averaged CO mixing ratios of 861 ppbv were recorded, which were about 200 ppbv higher than the daily averaged CO levels for April. Ozone mixing ratios during this period was about 9 ppbv higher than the daily averaged ozone levels for April. The spatial maps of MODIS fire counts during the period were also analyzed (Figure 14: top panel) and an increase in fire counts during the period was observed. Based on MODIS fire counts and $O_3$ and CO time series at bode ,we classified 20-25th March as pre-event period and April 3-6 and April 11-16 as high fire activity periods. During the first high fire period, an increase of 55% and 15% in surface ozone and CO mixing ratios, was observed. The fire maps during this period also indicate that the majority of this fire were occurring near Bode in Nepal region and is shown in orange color in spatial map (Figure 14: Top Panel). The AIRS satellite retrieved CO mixing ratios at 925hPa and 850hPa also indicate an increase in CO levels near Bode (indicated as black star), however no similar increase is observed in CO levels near Pantnagar (or IGP). The majority of air masses at 1km AGL prior to arriving to Bode (or Nainital also) are westerly and passes through IGP/northern Indian subcontinent in general. Since there is almost no fire activity near Punjab region so we argue that these nearby fire and their associated polluted plumes around Kathmandu

valley could affect the levels at Bode. Further, GFED biomass burning emissions were used to differentiate different types of biomass burning emissions during this period. During both of the events, the forest fire emissions dominated the total biomass burning emissions over Kathmandu region. The major land use types in black boxes indicated in Figure 14 are agricultural land or

forests. The meteorological data observed during this period did not show any noticeable differences and boundary layer height during these events was close to monthly average values. Therefore, we can conclude that forest fires occurring nearby Kathmandu valley were responsible for high levels of pollutants during this period.

The second event took place near first week of May when simultaneous increase in surface $O_3$ and CO levels at Bode, O3 levels at Nainital and CO levels at Pantnagar was observed. During this event, O3 mixing ratios at Bode and Nainital increased by 73% and 67% respectively. and CO mixing ratios at Bode and Pantnagar increased by about 24% and 58% respectively. The MODIS fire maps during these events show a large increase in MODIS fire counts in Punjab region (blue

box in top panel of Figure 15) in northwestern Indian subcontinent. During this period, total increase in fire counts is ~350% than that of April 25-30. The spatial distribution of AIRS retrieved CO mixing ratios at 925hPa and 850hPa, also indicate an overall increase in CO levels around northern part of Indian subcontinent. Similar increase is also observed in model (MOZART4/GEOS5) simulations where about two-fold increase in CO levels at 992 hPa during

this period is observed (Figure S7). Further, OMI tropospheric column $NO_2$ (30% cloud screened) also showed small enhancement during the same period (Figure S7). The fire emissions associated during this period indicate that majority of emissions are from crop residue burning unlike the previous case in April. Late April-early May is the harvest period for crops in northern India, and

wide spread crop residue burning is common during this period. This leads to release of massive amounts of pollutants over this region (Bhardwaj et al., 2016). The HYSPLIT back air trajectories also indicate that air masses during the period were arriving from these source regions to the observing sites. During this period, nothing noticeable in wind speeds, directions, temperature

solar radiation, and rainfall is observed so we could conclude that this increase in surface mixing ratios of $O_3$ and CO at these sites could be related to biomass burning in the northern Indian subcontinent. During this period, the influences are not only limited to surface level but also at higher altitudes where satellite retrieved vertical profiles of CO show high levels during this period. Rupakheti et al (2017) also reported a similar increase in the ambient concentrations of

BC, PM, CO and Ozone at Lumbini (regional site of SusKat campaign) during these two episodes were observed (7-9th April and 3-4th May). This site is located southwards of Kathmandu valley Himalayan foothills near IGP. The notable changes in surface $O_3$ and CO mixing ratios and GFED biomass burning emissions over Kathmandu region are shown in Table-4.

Similar  influences of wide spread biomass burning have also been observed in the vertical distribution of ozone over the central Himalayas. Details of balloon-borne observations of ozone (Ojha et al., 2014) and meteorological parameters along with inter-comparison of two kinds of meteorological sensors (i-Met and Vaisala) are given in Naja et al. (2016). The weekly balloon borne ozone profiles made from Nainital (on 9 May) also confirmed  an enhancement in ozone

(~16 ppb) in the lower troposphere (2-4 km) when compared with the ozone profile on 1 May (Figure 16). The enhancement is about 14 ppbv in 4-6 km region. Such events are generally observed during the spring season, when the influence of regionally polluted air masses from the IGP could travel over long distances.

**3.8 Influences of Stratosphere-Troposphere exchange (STE) on surface ozone levels at Bode**

The tropopause folding events and influences of STE over the Indian subcontinent and Tibetan plateau are more frequent during winter and early spring seasons (Cristofanelli et al., 2010; Chen et al., 2011; Phanikumar et al., 2017) and a few attempts in the past were made to understand the role of STE over the Indian subcontinent (Mandal et al., 1998; Ganguly and Tzanis, 2011). Here, to understand the role of STE on surface ozone levels at Kathmandu vertical distribution of ertel potential vorticity (EPV) using MERRA v2.0 reanalysis, AIRS satellite retrieved ozone, relative humidity (RH) and CO were observed during Jan-May 2013 (Figure 17). The EPV distribution is represented in potential vorticity unit threshold (EPV $> 1.6$ PVU $= 1.6$ x $10^{-6}$ K m$^2$/Kg/sec) defined by Cristofanelli et al., (2006). Since the EPV near extratropical tropopause is at about 2 PVU and EPV in the stratosphere is about 1-2 magnitude higher, therefore, any values of EPV greater than 1.6 are suggested to be associated with the downward transport of ozone rich air masses from above (Cristofanelli et al., 2006). Overall, EPV distribution suggests downward transport associated with STE is limited to upper and middle part of the troposphere (Top Panel: Figure 17). The ozone distribution also exhibit similar behavior where increases in ozone levels (downward transport) reaching surface are negligible. However, a seasonal increase in ozone mixing ratios at lower and middle troposphere can be observed during Jan to May. To investigate the role of STE during April and May vertical profiles of EPV, O$_3$, CO and RH were studied for these periods (Figure 18). During April event (Fig. 18, Top Panel) EPV during April 11-16 was found to be increased as compared to prevent period (March) but it was reduced during April 3-6 period. During both the high fire periods ozone, CO were higher during the event which we propose is due to biomass burning since during STE reductions in CO are also observed. So therefore April

event do not show any clear evidences of STE based on our analysis. Further, during May (Fig. 18, Bottom Panel) an increase in EPV at higher altitudes and $O_3$ at lower altitude is observed but again CO didn't showed any decrease during this period. Therefore, we conclude that during both of these high pollution periods, no signs of downward transport is observed.

## 4. Summary

This study provides information about the regional distribution of $O_3$ and CO during the SusKat-ABC field campaign (Jan-Jun 2013) by analyzing simultaneous surface measurements of ozone and CO from Bode in the Kathmandu Valley and from two Indian sites, Nainital and Pantnagar.

Results from few air samples and their analysis for eight ($C_2$-$C_5$) light non-methane hydrocarbons are also presented. The diurnal variations show higher levels of ozone during daytime and morning/evening peaks in CO. This daytime build-up in ozone is consistent during all months, with a relatively smaller increment during the month of June due to prevailing cloudy or rainy conditions. Such a daytime increase in surface ozone is mainly due to photochemical production

from precursor gases in the presence of sunlight. Very low nighttime levels of ozone were also observed during the winter season, which can be attributed to the titration of $O_3$ by NO. The diurnal variations in CO showed two peaks during morning and evening hours, due mainly to rush hour traffic sources and cooking activities, and a similar distribution is also observed in Pantnagar. The evening peak was relatively less prominent at Bode, due to fast westerly winds blowing across the

valley during afternoon that flush out CO in contrast to calm nighttime winds and a shallow nocturnal boundary layer, resulting in the highest levels during morning time. After reaching its maximum levels during morning time (up to 2300 ppbv in winter months), the levels decrease as

the day progresses. This decrease is attributed to the boundary layer evolution and strong winds blowing across the Valley which dilutes the CO levels.

The correlations between $O_3$ and CO are found to be negative in the winter period ($r^2$=0.82 in January and $r^2$=0.71 in February) and this negative correlation becomes weaker gradually, with the lowest value in May ($r^2$=0.12). Hourly average CO levels also show a systematic decrease from its level of about 2100 ppbv in January to about 600 ppbv in June, whereas ozone shows the opposite tendency. A weaker negative correlation is observed during early morning (0300-0500 and 0730-0830 hours) or nighttime hours (2200-2300 hours) while a slight positive correlation is seen during the noon period (1300-1500 hours). The background $O_3$ and CO levels at Bode are found to be about 14 ppbv and 325 ppbv respectively. It is shown that $O_3$ , CO and light non-methane hydrocarbon levels are higher at Bode than those at the IGP site (Pantnagar) analyzed here and in the Himalayan foothills, particularly in winter. The rate of change of ozone during morning and evening hours is different at Bode, with a faster ozone increase rate during the day (about 17 ppbv/hour) but a slower ozone decrease rate (5-6 ppbv/hour) in the evening, suggesting the prevalence of a semi-urban kind of environment at Bode.

During the spring season, northern Indian biomass burning is found to affect the measurements sites in both India and Nepal. Two distinct events of biomass burning influence corresponding to the first half of April and May 2013 were studied. During both of these periods, an increase in $O_3$ and CO is observed over Bode. A similar increase is also observed at Nainital during May, but not

during April. During first week of April, a sharp increase of (~200 ppbv) in average CO mixing ratios is observed at Bode and increase is also observed in ozone levels. The analysis of spatial distribution of MODIS retrieved active fire locations indicated that the majority of fires took place in the nearby Nepalese regions that are upwind of Bode but downwind of Nainital. Analysis of

biomass burning emission inventories indicated these emissions originated primarily from burning of the forests. Satellite retrievals of lower atmospheric CO mixing ratios also indicated an increase in CO levels during the event period near Bode region but not near Pantnagar and Nainital. During the first week of May, simultaneous increases in $O_3$ and CO levels were observed at Bode, and Nainital. The MOZART simulations during that period also indicate about a two-fold increase in

near-surface CO levels. The MODIS-derived fire location showed a ~256% increment over the Punjab region in the IGP. Analysis of biomass burning emission inventories indicated that fires during May 2013 originated mainly from the crop residue burning.

Analysis of back-air trajectories showed that majority of the air masses passed over the fires in Punjab before arriving at Nainital, Pantnagar and Bode. Similar increases in near-surface

distribution of satellite retrieved CO mixing ratios around all the sites are also observed. The balloon borne ozone profiles from Nainital also confirmed the significant enhancement in ozone (~16 ppbv) in the lower troposphere between the balloon flights on May 1[st] and 9[th], 2013. Such events are mainly observed during the spring season when the influence of regionally polluted air masses from the IGP region are observed over measurement sites in the Himalayan region. This

study provided the first regional picture of the air quality during the SusKat field campaign. Future studies should focus on long-term continuous and collocated $NO_x$ and NMHCs measurements to advance our understanding of atmospheric chemistry in this region. The SusKat dataset must also be used to identify and address the weaknesses of state-of-the-science air quality models and

emission inventories, which together with observations will play an important role in developing effective mitigation strategies for this region and ultimately reducing the vulnerability of public health to frequently occurring air pollution episodes.

5  **Acknowledgments**

SusKat field campaign was supported by IASS, Germany and ICIMOD, Nepal. $O_3$, CO and light NMHCs observations are supported by ATCTM project of ISRO-Geosphere Biosphere Program and ARIES, DST. Authors are thankful to Bhogendra Kathayat, Dipesh Rupakheti and Shyam Newar for their help in conducting observations at Bode. The IASS is supported by the German Ministry of Education and Research (BMBF) and the Brandenburg State Ministry of Science, Research and Culture (MWFK). The National Center for Atmospheric Research is sponsored by the National Science Foundation. We are grateful to teams of OMI, AIRS, MERRA-2, GFED, MOZART and HYSPLIT for making available the respective data. We are thankful to two anonymous reviewers for their fruitful comments.

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

| Month | Ozone (ppbv) | Max/Min (ppbv) | Daytime average ozone ppbv (1100-1700 hours) | CO (ppbv) | Max/Min (ppbv) |
|---|---|---|---|---|---|
| Jan | 23.5 ± 19.9 | 87.1/1.4 | 49.8 ± 10.2 | 832 ± 422 | 2323/218 |
| Feb | 25.6 ± 20.4 | 95/1.2 | 49.9 ± 13.9 | 717 ± 397 | 2182/162 |
| Mar | 37.4 ± 23 | 105.9/1.2 | 61.8 ± 12.0 | 698 ± 364 | 2011/158 |
| Apr | 43.5 ± 26.6 | 116.2/1.4 | 67.0 ± 20.4 | 667 ± 372 | 1969/175 |
| May | 38.6 ± 21.4 | 111.1/1.9 | 55.1 ± 18.9 | 401 ± 213 | 1656/146 |
| Jun | 31.1 ± 16 | 68.4/1.7 | 46.5 ± 8.5 | 303 ± 85 | 676/166 |

**Table 2:** Average (avg), standard deviation (std), maximum (max), minimum (min) and daily counts of $O_3$ and CO values during four time periods for the entire observational period (January – June 2013).

| Time period | | 0300-0500 hr | 0730-0830 hr | 1300-1500 hr | 2200-2300 hr |
|---|---|---|---|---|---|
| Ozone | Avg (ppbv) | 13.1 | 13.9 | 58.9 | 27.8 |
| | Std (ppbv) | 1.2 | 1.7 | 10.0 | 7 |
| | Max (ppbv) | 54.0 | 52.5 | 102.4 | 70.8 |
| | Min (ppbv) | 1.8 | 2.0 | 25.9 | 1.4 |
| | Counts | 158 | 158 | 158 | 158 |
| CO | Avg (ppbv) | 833.8 | 1103.2 | 325.4 | 626.1 |
| | Std (ppbv) | 292.6 | 380.3 | 98.3 | 306 |
| | Max (ppbv) | 1770.0 | 2430.0 | 910.0 | 1820.0 |
| | Min (ppbv) | 150.0 | 160.0 | 160.0 | 190.0 |
| | Counts | 159 | 158 | 159 | 158 |

**Table 3:** Average, standard deviation, minimum and maximum mixing ratios (ppbv, except for $CH_4$, which is in ppmv) of $CH_4$, CO and eight ($C_2$-$C_5$) light-NMHCs from the analysis of daily air

14    sample collection from 30 December 2012 to 14 January 2013. Percentage contribution for each

15    NMHCs to the total of measured NMHCs is also given.

| Gases | Average | Standard deviation | Minimum | Maximum | % Contri bution | Analysis |
|-------|---------|--------------------|---------|---------|-----------------|----------|
| Methane | 2.55 | 0.12 | 2.39 | 2.87 | -- | 15 |
| CO | 392.5 | 109.3 | 272 | 588.8 | -- | 16 |
| Ethane | 3.49 | 1.24 | 1.01 | 6.35 | 15.8 | 15 |
| Ethene | 2.84 | 2.37 | 0.31 | 9.69 | 12.9 | 15 |
| Propane | 4.41 | 4.14 | 0.44 | 15.48 | 20.0 | 13 |
| Propene | 1.06 | 0.91 | 0.28 | 3.86 | 4.8 | 13 |
| i-butane | 2.26 | 1.93 | 0.24 | 7.78 | 10.3 | 13 |
| Acetylene | 4.08 | 3.87 | 0.34 | 14.35 | 13.5 | 14 |
| n-butane | 2.96 | 1.80 | 0.18 | 5.81 | 18.5 | 14 |
| i-Pentane | 0.92 | 0.84 | 0.15 | 2.63 | 4.2 | 14 |

24 **Table 4:** The average $O_3$ and CO mixing ratios at Bode, Nainital, Pantnagar with GFED average

biomass burning emissions over Kathmandu region during different periods.

| Fire Periods | Fire Count | Ozone (ppbv) | | CO (ppbv) | | Avg. Biomass burning emissions (Tg/day) | | |
|---|---|---|---|---|---|---|---|---|
| | | Bode | NTL | Bode | PNT | Total | Crops | Forest |
| LFAP (Mar 1-Mar 31) | 5 | 37.4 | 45.2 | 705 | 455 | 8.96 | 2.29 | 5.87 |
| HFAP (Apr-May) | 70 | 43.7 | 63.9 | 504 | 374 | 63.70 | 42.70 | 19.94 |
| | | | | | | | | |
| Mar (20-25) | 3 | 34.8 | 46.1 | 693 | 401 | 7.17 | 2.37 | 3.67 |
| Apr (3-6) | 18 | 46.7 | 49.2 | 797 | 250 | 57.74 | 7.48 | 46.46 |
| Apr (11-16) | 27 | 54.0 | 57.9 | 762 | 265 | 52.38 | 7.03 | 44.16 |
| | | | | | | | | |
| Apr (26-30) | 26 | 33.9 | 45.0 | 505 | 313 | 22.10 | 14.60 | 6.78 |
| May (2-6) | 116 | 58.8 | 74.8 | 625 | 495 | 128.40 | 97.58 | 29.72 |

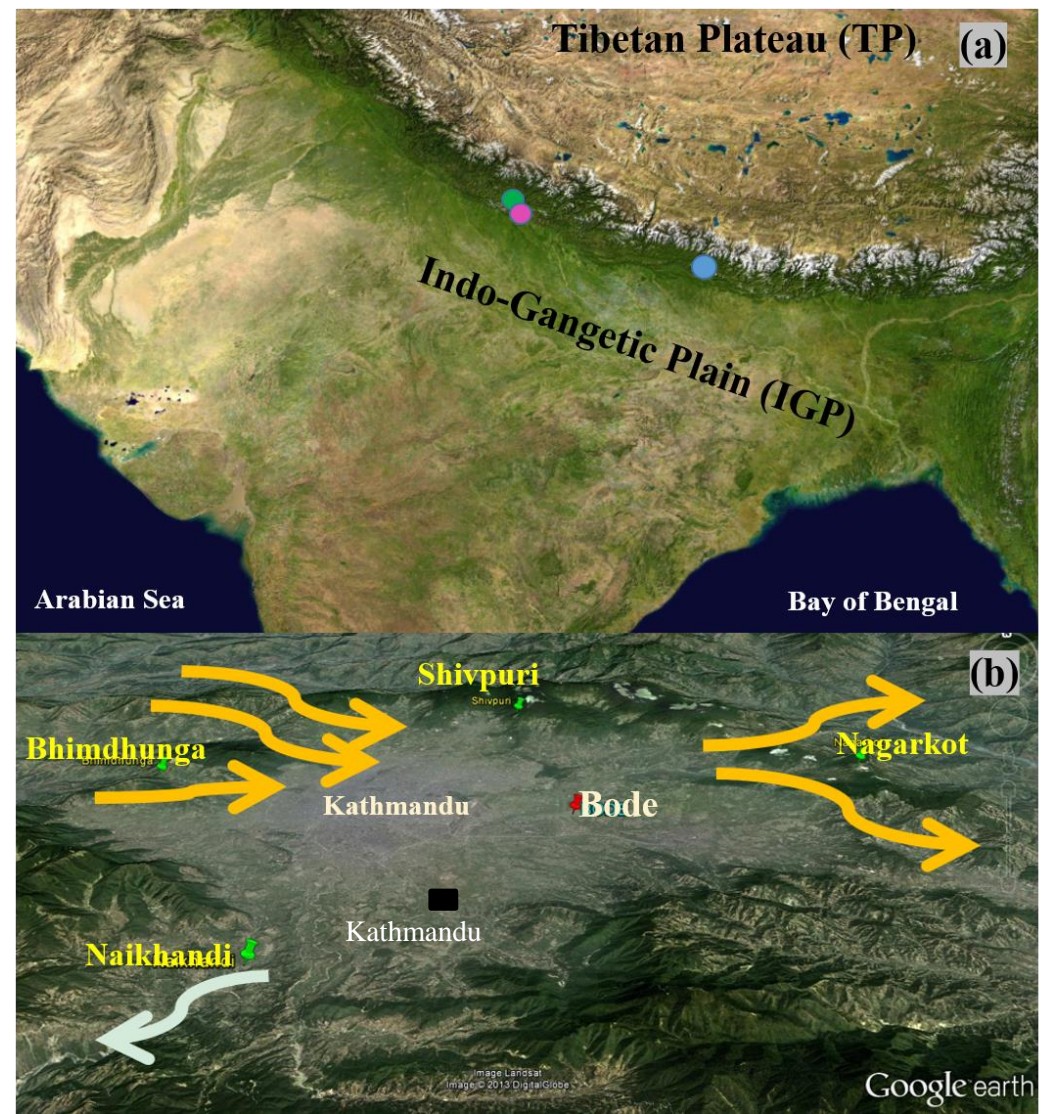

**Figure 1:** (a) Satellite image depicting the location of observation sites *viz.,* Bode (Blue) in Kathmandu Valley, Nepal and Nainital (Green) and Pantnagar (Pink) in India during SusKat field campaign. (b) Satellite image of the Kathmandu Valley (edge on view) with the super site (Bode) and 4 satellite sites (Bhimdhunga, Naikhandi, Nagarkot, and Shivpuri). Figure (b) also indicates the position of five mountain passes surrounding the valley (yellow arrows) and one river outflow location (white). Kathmandu city is also marked by a black square.

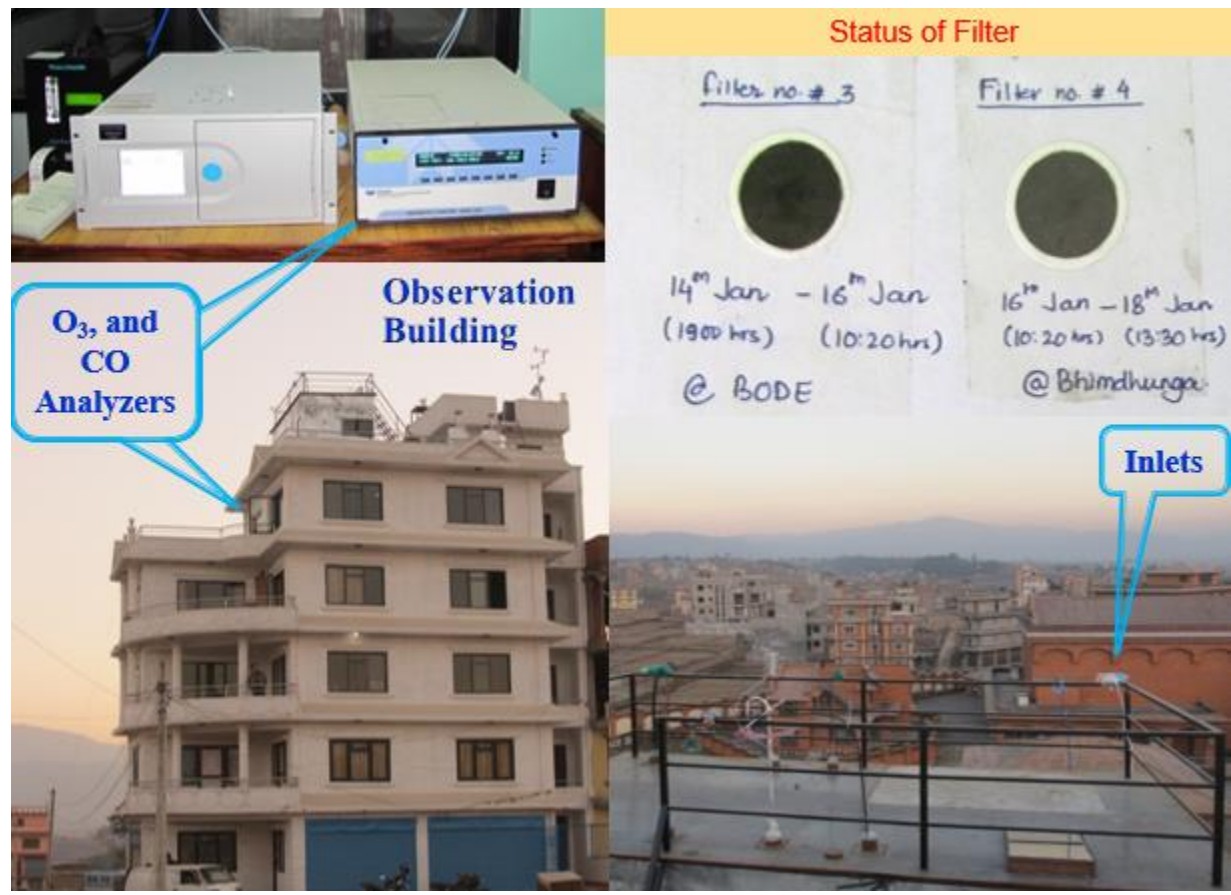

**Figure 2:** The observation setup with $O_3$ and CO analyzers (top-left) placed at the fourth floor in

a building at Bode, Nepal (bottom-left). The position of sampling inlets are towards eastern side

of the valley (bottom-right). Almost black inlet filters are seen in about 2 days of operations at

Bode and Bhimdhunga sites (top-right).

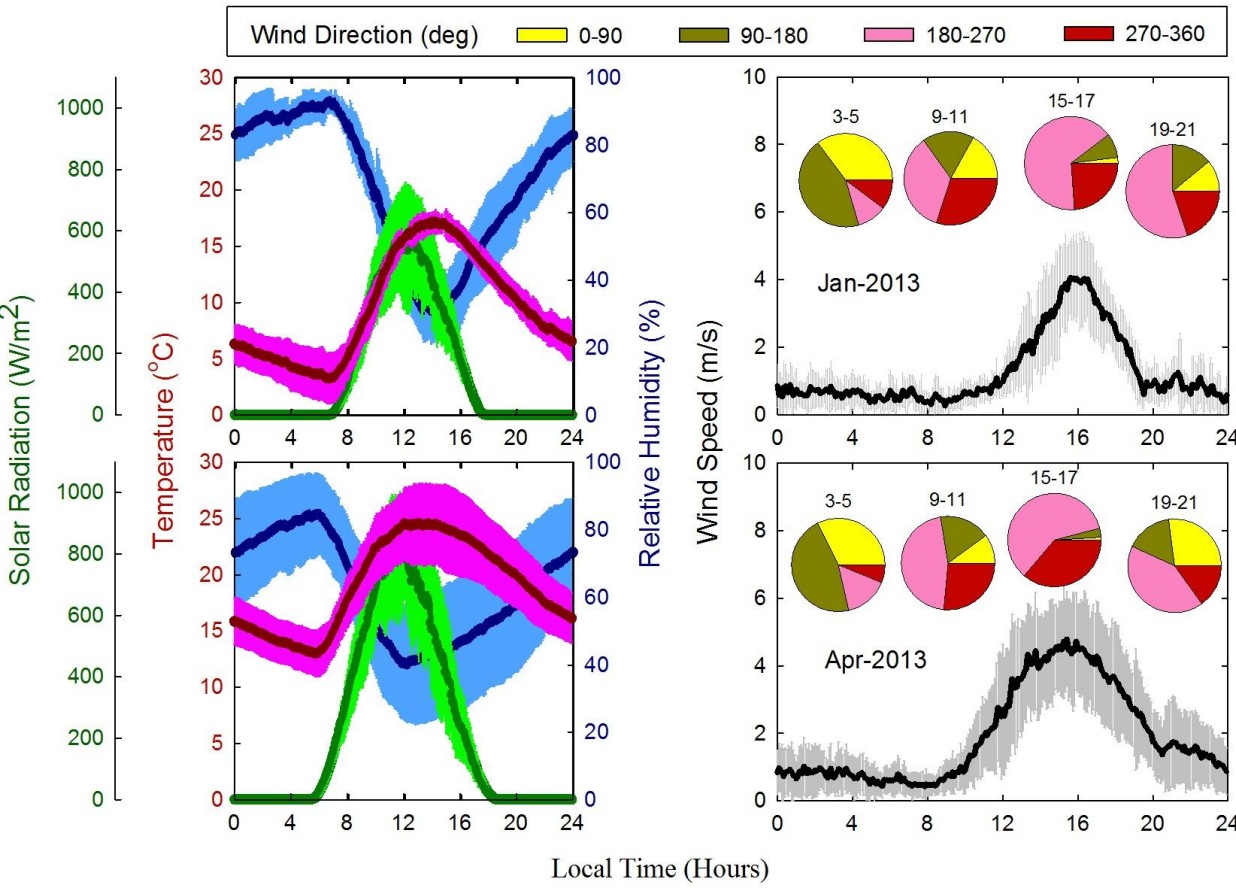

**Figure 3:** Average diurnal variations in temperature, solar radiation, relative humidity, wind speed
and direction at Bode during January (upper panel) and April 2013 (bottom panel). These two
months are taken as representative for winter and spring season respectively. (right) Wind
directions during four time periods (0300-0500, 0900-1100, 1500-1700 and 1900-2100 hours) are
shown as pie charts in wind speed plots.

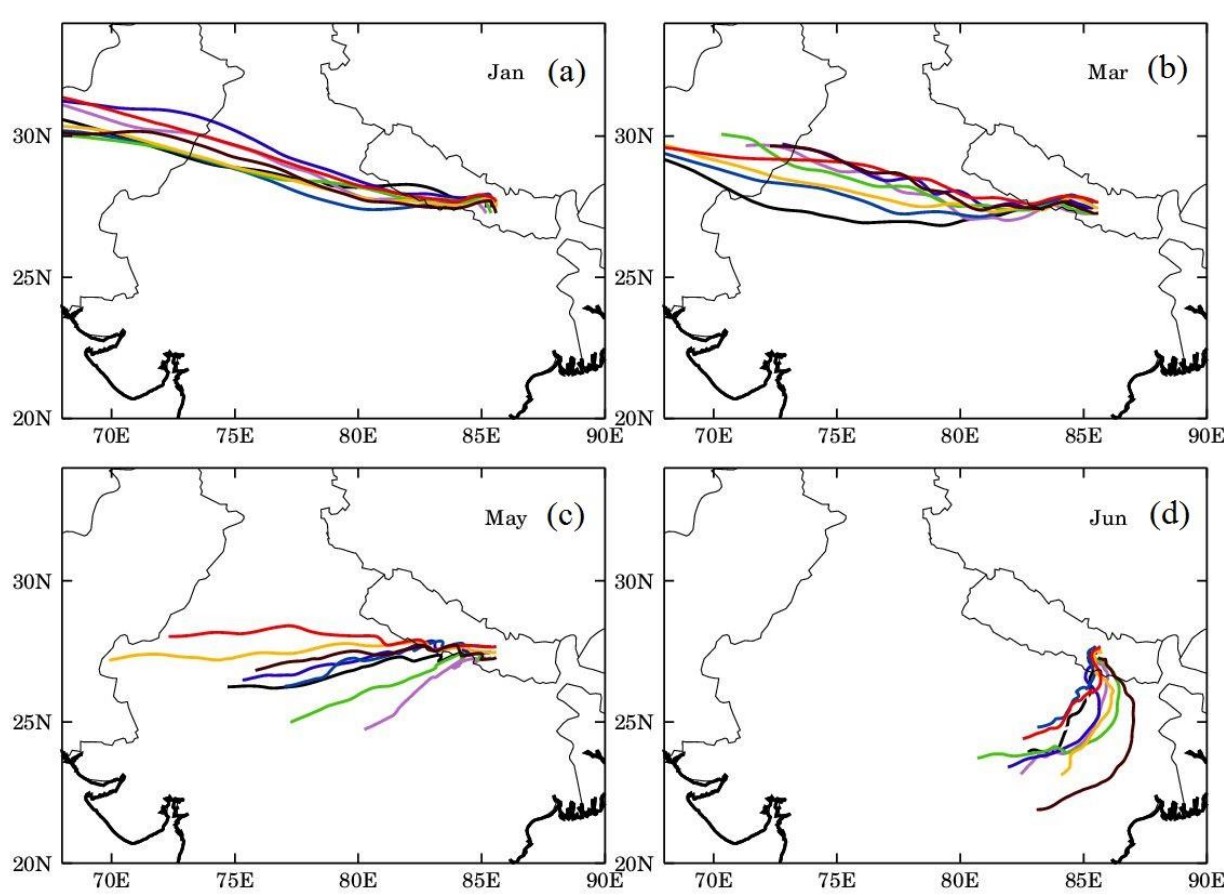

**Figure 4:** Five days nine particles HYSPLIT back-air trajectories over Bode region during (a)

January, (b) March, (c) May and (d) June. The colored trajectories are of monthly averaged for

each nine particles during the respective months.

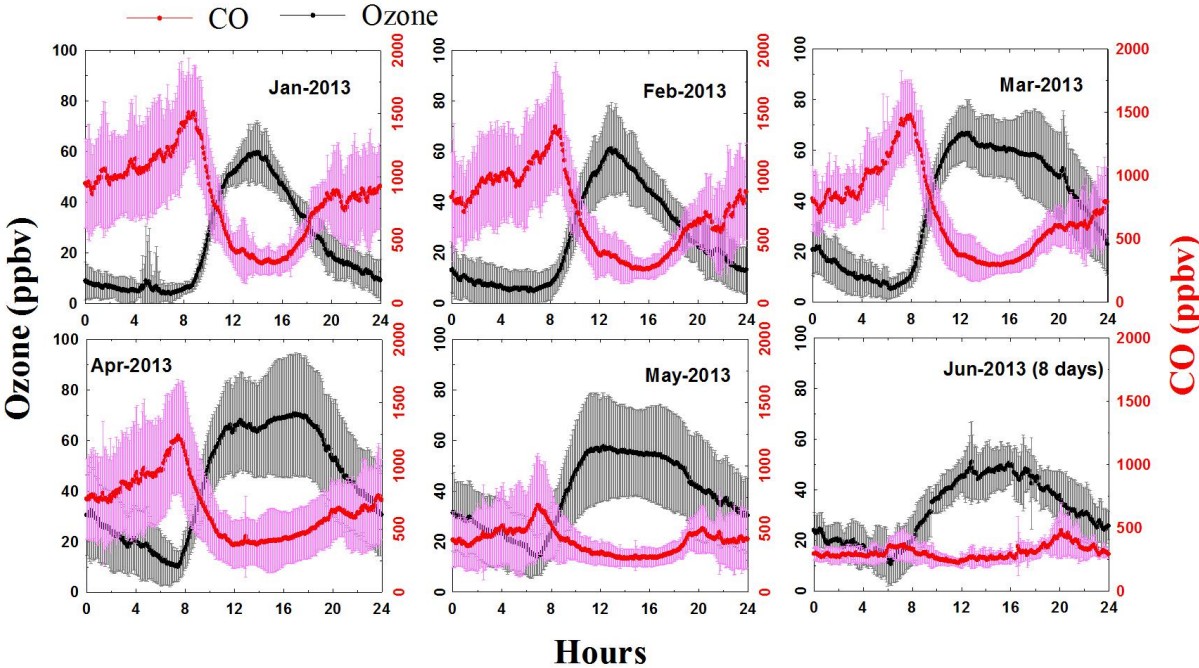

**Figure 5:** Monthly variations in average diurnal $O_3$ and CO mixing ratios with 1-sigma spread at

56 Bode during January-June 2013.

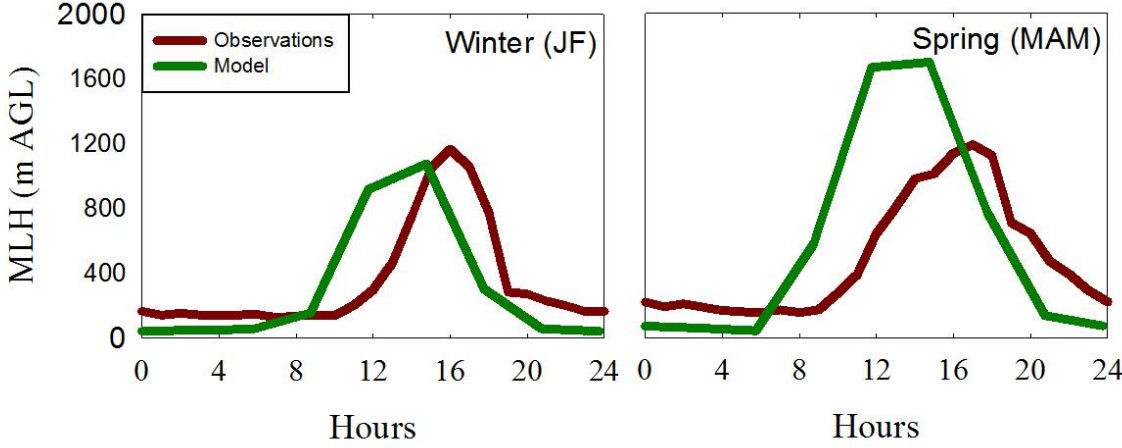

**Figure 6**: Observed (red) and model (green) simulated boundary layer height during winter and

60 spring at Bode.

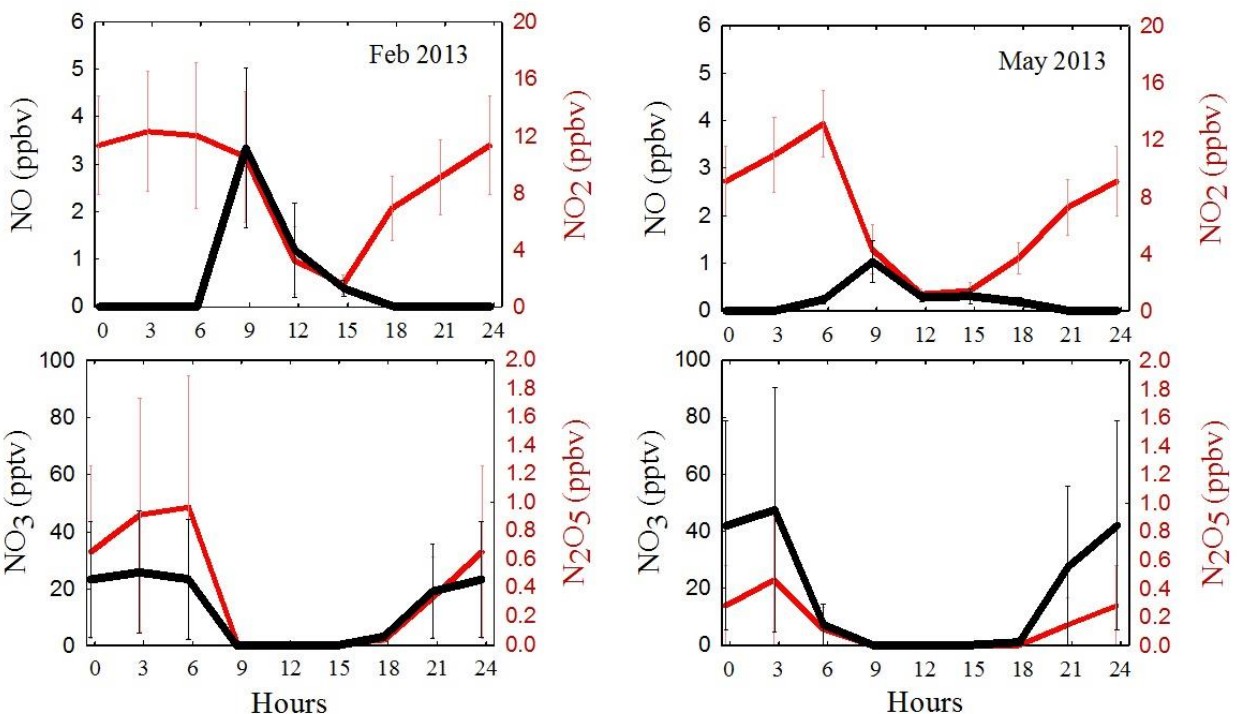

**Figure 7**: Model simulated average diurnal variations in NO, $NO_2$, $NO_3$ and $N_2O_5$ during February and May 2013.

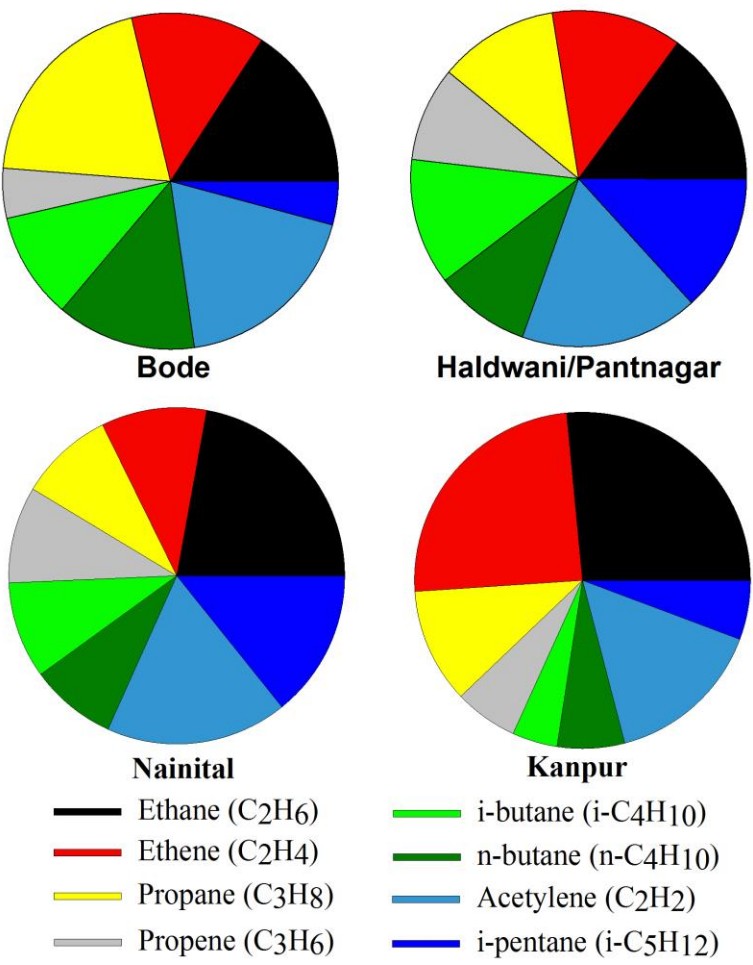

**Figure 8:** Contribution of eight light non-methane hydrocarbons (NMHCs) to the total NMHCs at Bode, Pantnagar/Haldwani, Nainital and Kanpur in December. Bode includes January data too.

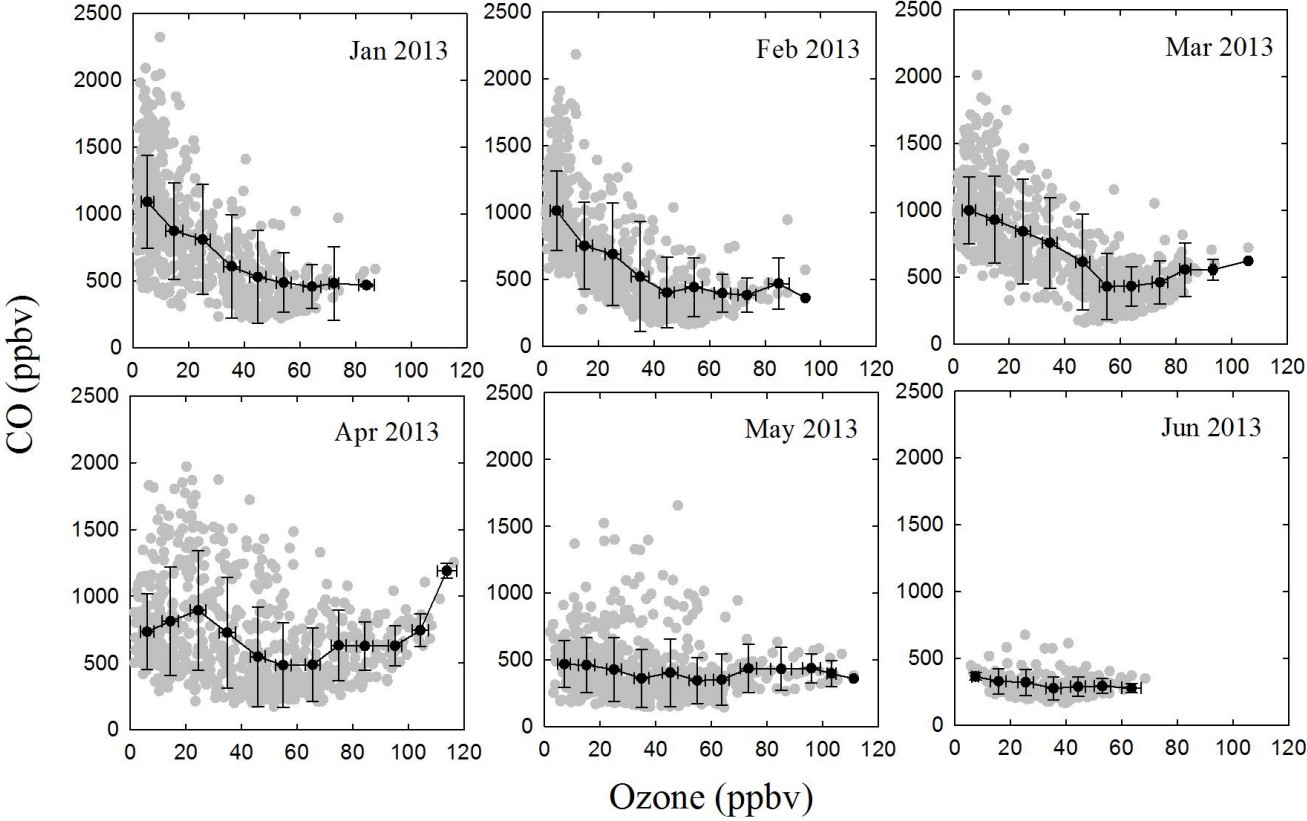

**Figure 9:** Relation between ozone and CO from January 2013 to June 2013 at Bode. Grey dots

are hourly average data and black filled dots are 10 ppbv binned averaged with respect to ozone.

The spread around the mean value is one sigma value.

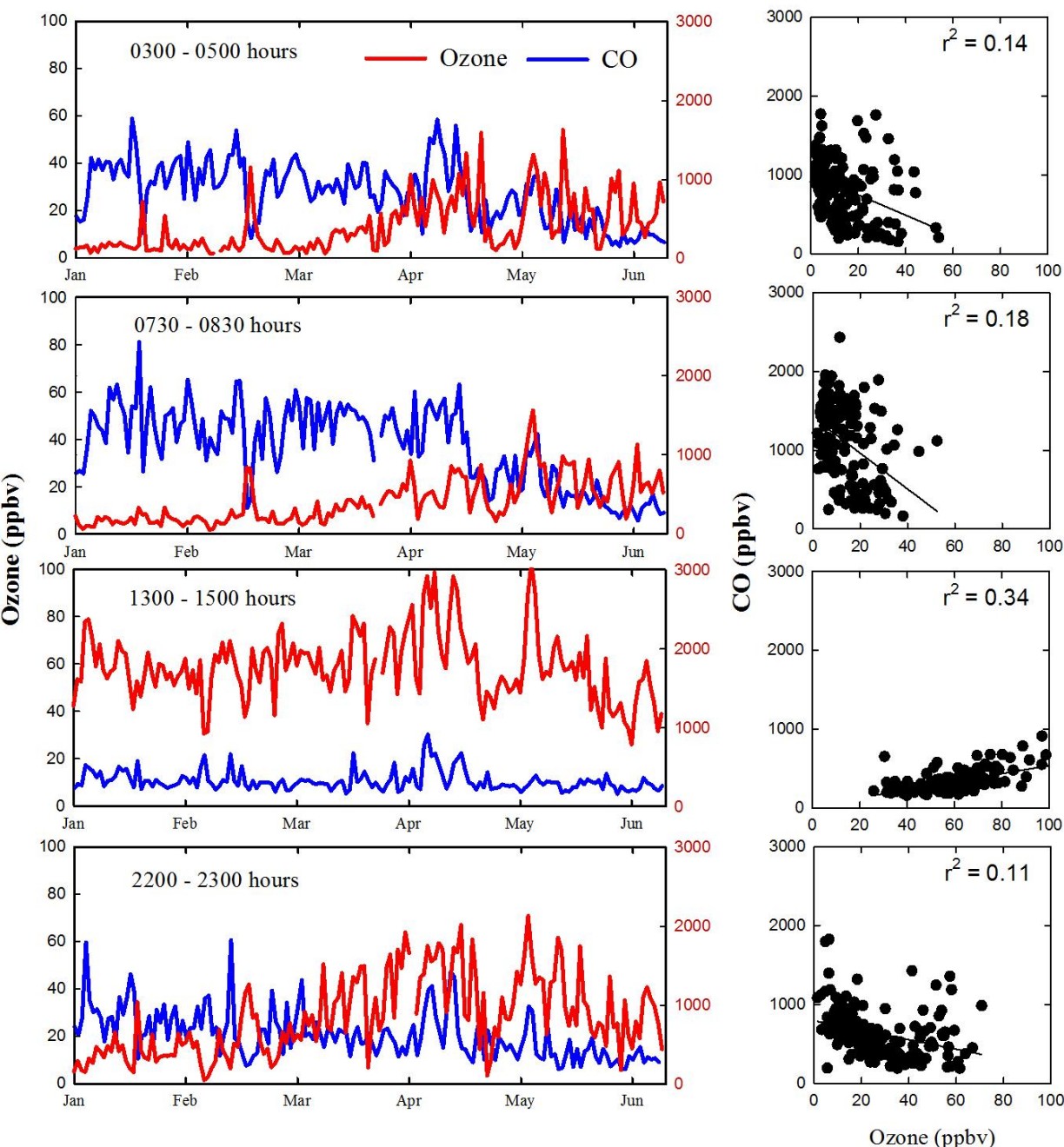

**Figure 10:** Variations in ozone and CO during four-time periods at Bode, Nepal. Correlation

between them is also shown (right).

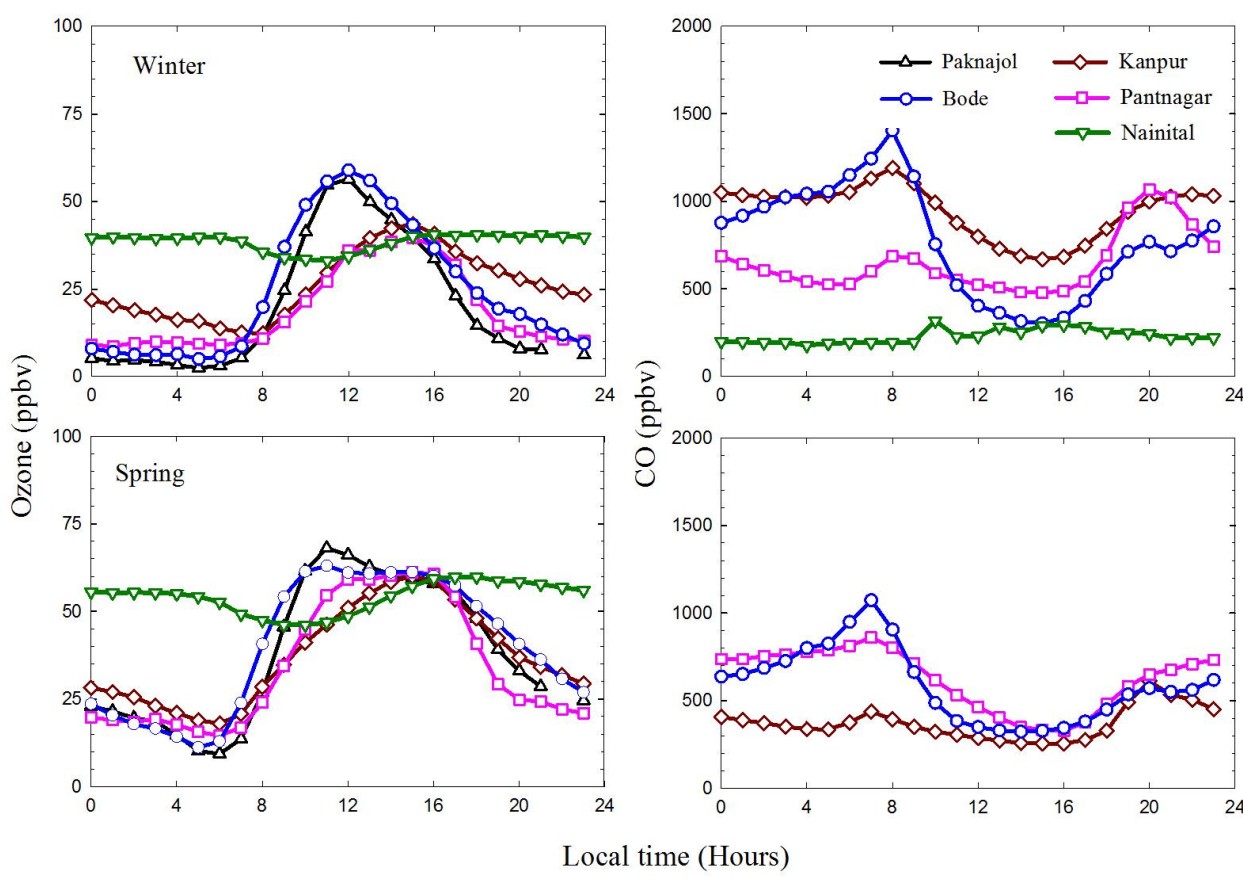

**Figure 11:** Averaged diurnal variations (winter on top and spring at bottom) in ozone and CO at

Bode, Pantnagar, Nainital. Surface ozone observations at Kanpur (data from Gaur et al., 2014) and

Paknajol (data from Putero et al., 2015) are also shown for the comparison. CO measurements

were not available at Nainital during the spring season.

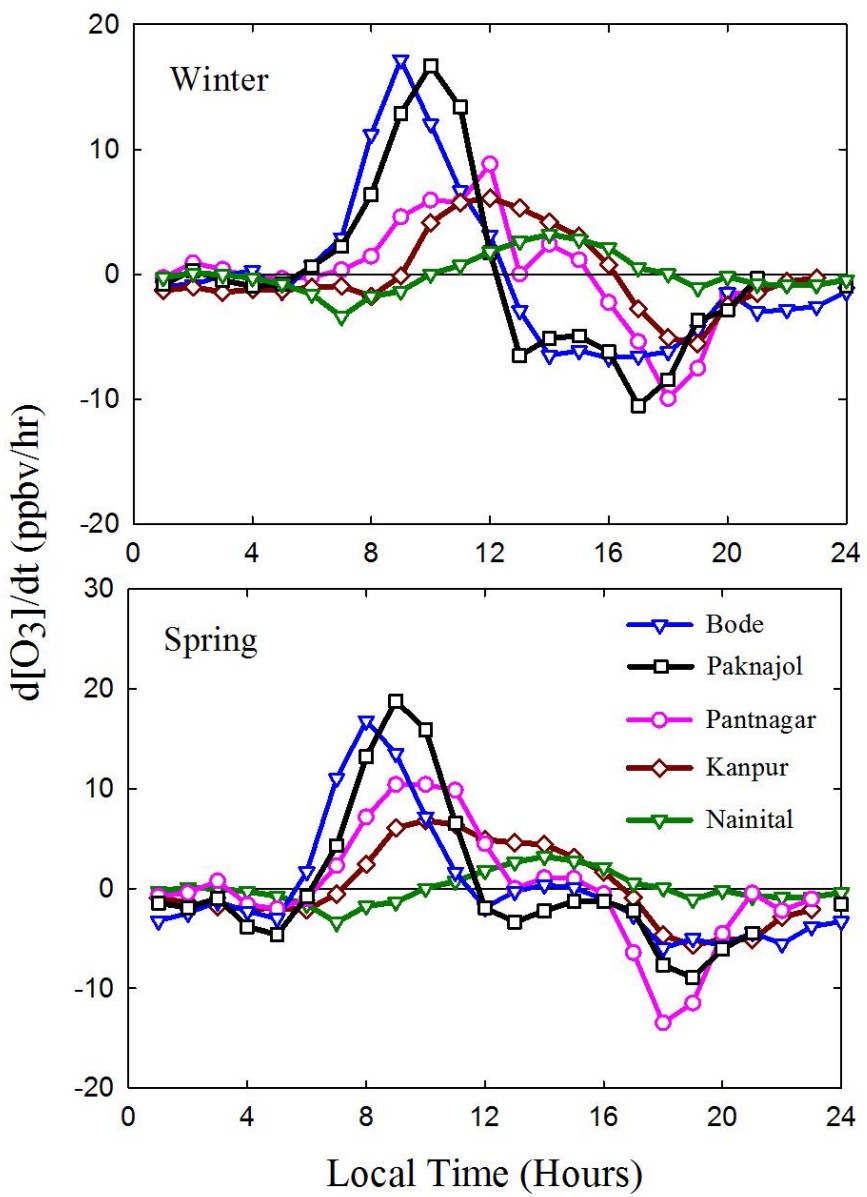

**Figure 12:** Diurnal varitions in the average rate of change of ozone during winter and spring 2013 at Bode (blue), Paknajol (black; Putero et al., 2015), Pantnagar (Pink), Kanpur (brown; Gaur et al., 2014) and Nainital (green).

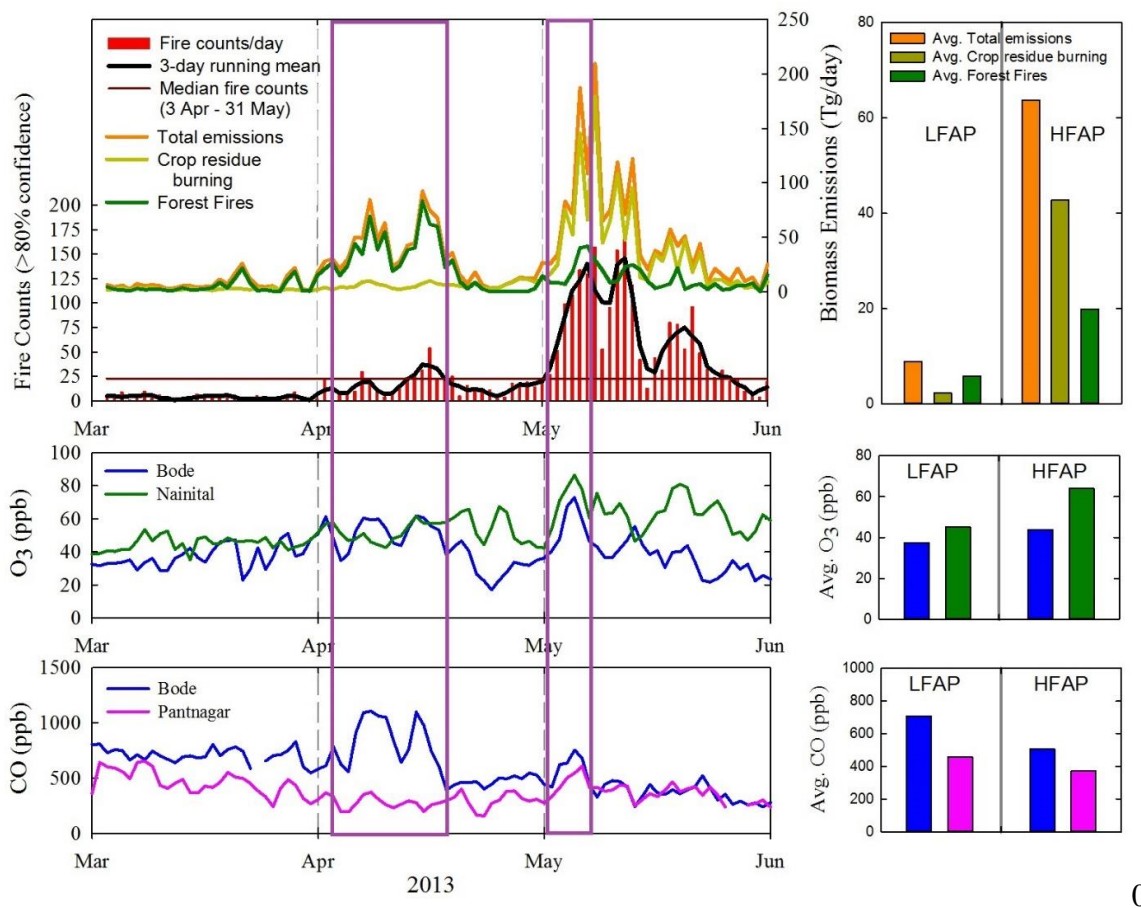

**Figure 13:** Top Left: time series of MODIS daily fire counts (red bar), 3-day running mean (black lines), median fire counts (brown line) for the fire period (3 April31 May 2013). Total biomass burning emissions (orange line), crop residue burning emissions (dark yellow line) and forest fire emissions (dark green) over 1°x1° grid box around bode (27-28°N, 85-86°E) are also shown. Top right: average biomass burning emissions for two fire activity periods over Bode region. Center: time series of surface ozone mixing ratios at Bode and Nainital (line plot-Left) and average ozone mixing ratios during two fire periods (as bar plot-Right), respectively. Bottom: time series of surface CO mixing ratios at Bode and Nainital (line plot-Left) and average CO mixing ratios during two fire periods (as bar plot-Right), respectively. The two fire events in April and May are also shown highlighted (in violet boxes).

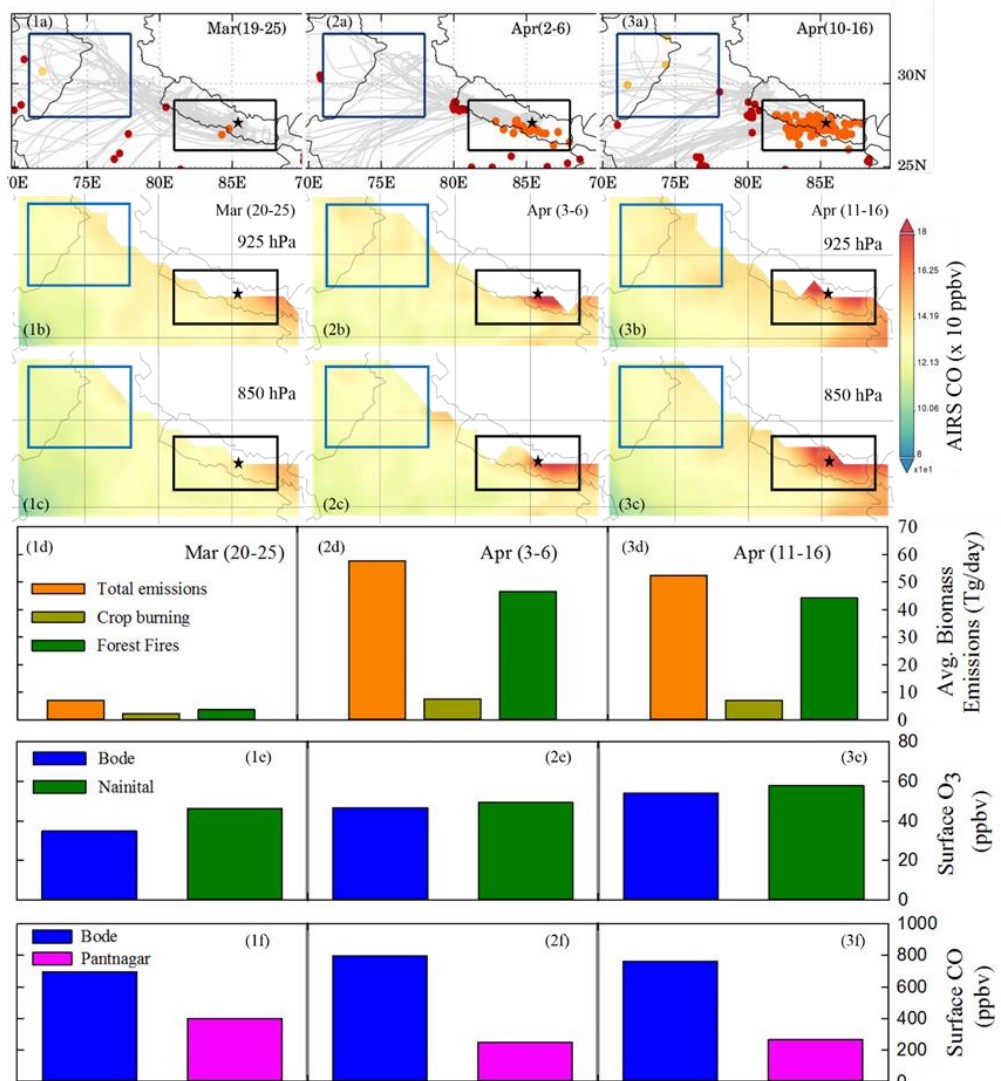

**Figure 14:** Top (1a-3a): Spatial distribution of MODIS fire counts over Northern Indian subcontinent during three periods (left to right). The black boxes represent two fire hotspots (refer Fig. S6) over the region with fire counts shown as different colors during the three periods. Center (1b-3c): spatial distribution of AIRS CO mixing ratio at 925hPa (1b-3b) and 850 hPa (1c-3c) during three periods (left to right). Center (1d-3d): Average biomass burning emissions over Bode region (27-8oE, 85-86oE) using GFED v4.0 inventory during the three periods (left to right). Bottom Panels (1e-3f): changes in average surface mixing ratios of O3 (1e-3e) and CO (1f-3f) at different sites during the three periods (left to right).

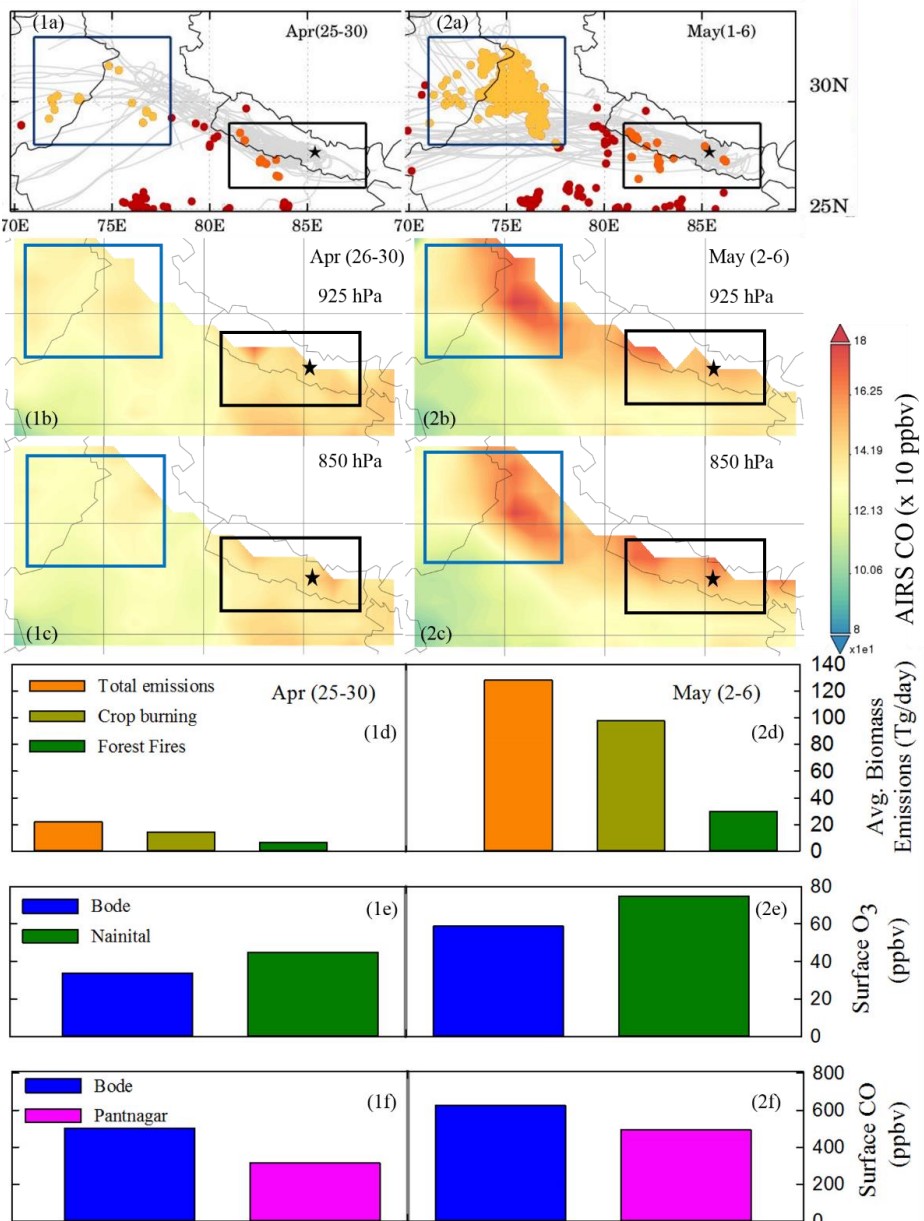

**Figure 15:** Top (1a-2b): Spatial distribution of MODIS fire counts over Northern Indian subcontinent during two periods (left to right). The black boxes represent two fire hotspots (refer Fig. S1) over the region with fire counts shown as different colors during different periods. Center (1b-2c): spatial distribution of AIRS CO mixing ratio at 925hPa (1b-2b) and 850 hPa (1c-2c) during two periods (left to right). Center (1d-2d): Average biomass burning emissions over Bode

region (27-8ºE, 85-86ºE) using GFED v4.0 inventory during the two periods (left to right). Bottom

Panels (1e-2f): changes in average surface mixing ratios of O₃ (1e-2e) and CO (1f-2f) at different

sites during the two periods (left to right).

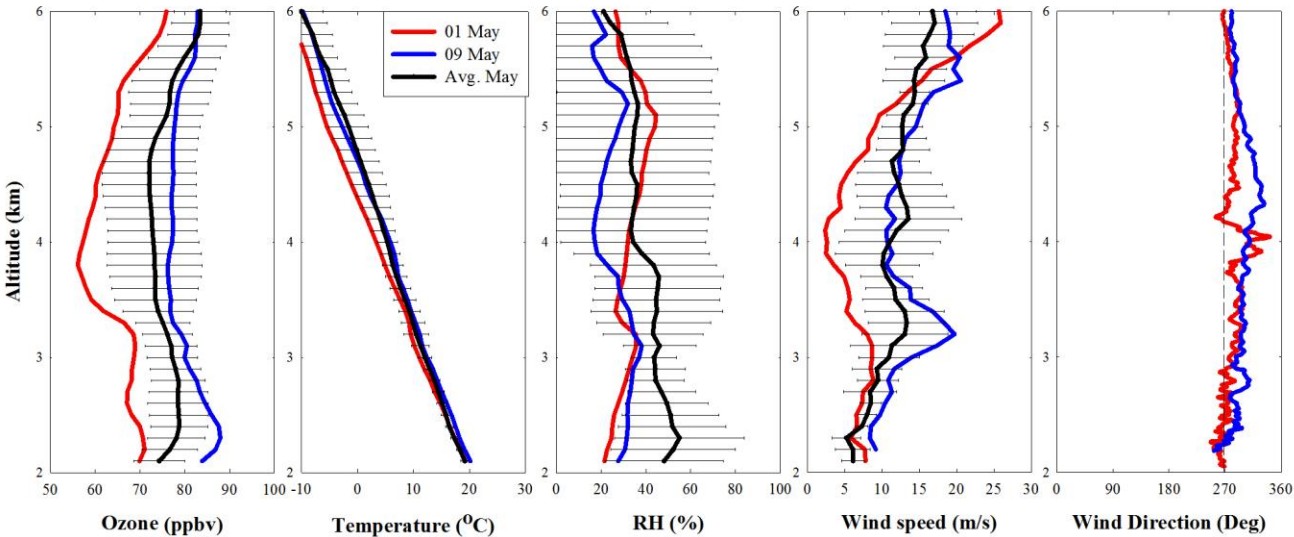

**Figure 16:** The vertical profiles of ozone, temperature, RH, wind speed and direction over Nainital

region on 1st (Red) and 9th (Blue) May 2013. The black lines show respective monthly average

(May 2013) vertical profiles with bars representing one-sigma variations.

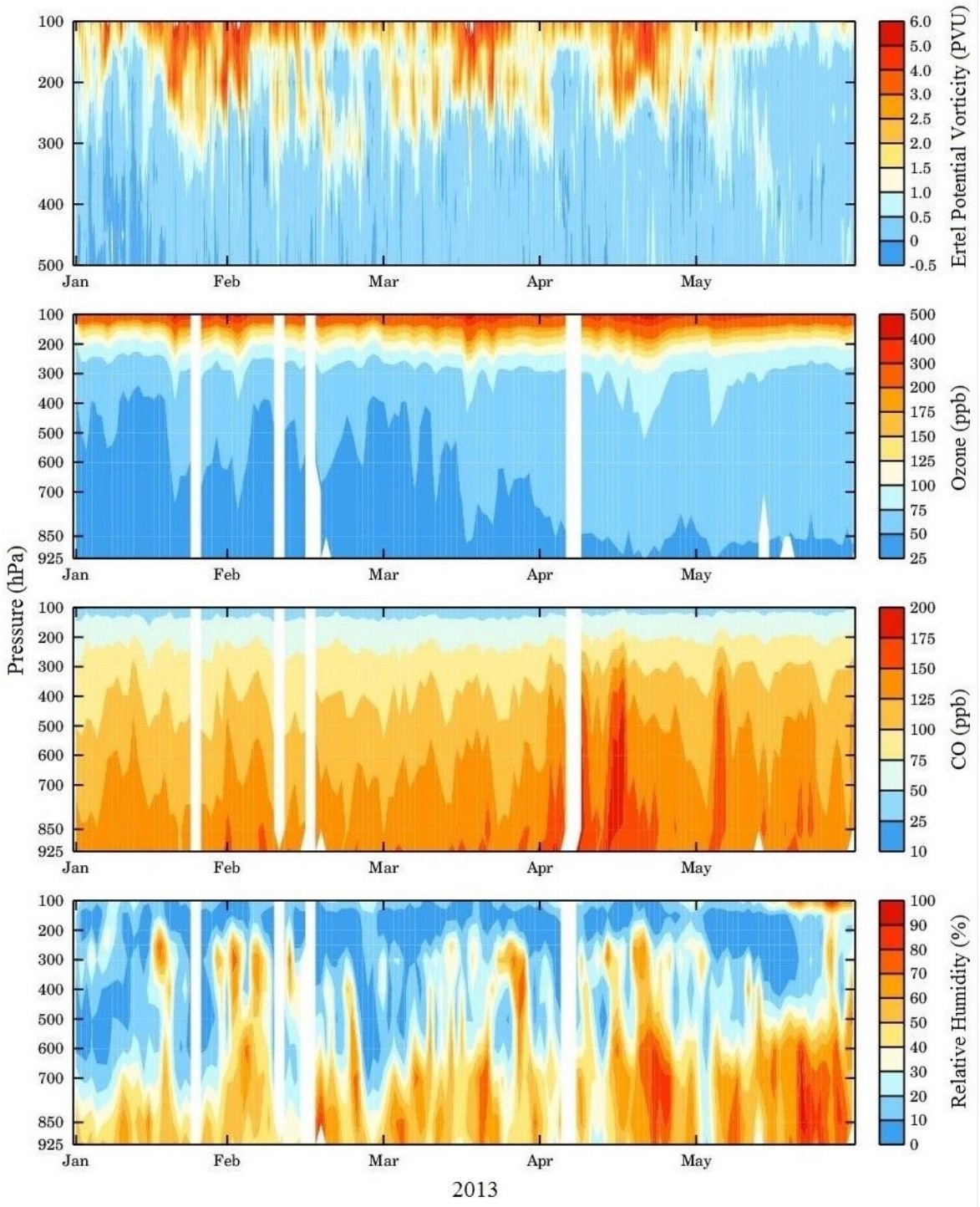

**Figure 17:** Vertical distribution of ertel potential vorticity (EPV) calculated by MERRA v2
reanalysis (top), AIRS retrieved ozone mixing ratios (centre), CO mixing ratios (center) and
relative humidity (RH) during Jan-May 2013.

143

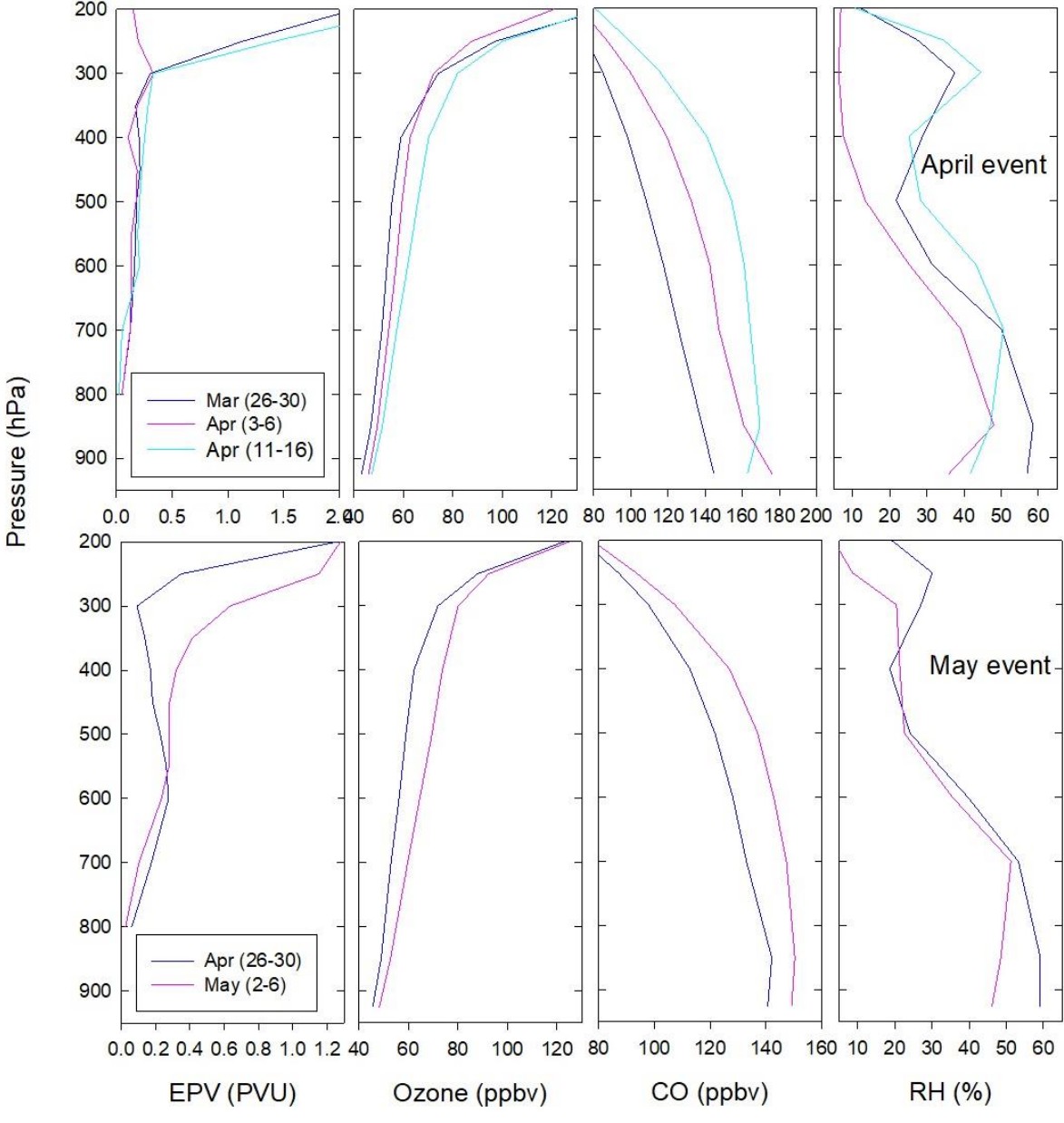

Figure 18: Vertical profiles of EPV, ozone, CO and RH during April (Top Panels) and May (Bottom Panels) biomass burning episodes.