# Peer review of "Variations in surface ozone and carbon monoxide in the Kathmandu Valley and surrounding"

_Atmospheric Chemistry and Physics, 2017_

## Referee Comment (RC1) · Anonymous Referee #2 · 9 Aug 2017

The paper presents data collected during the winter and spring of 202-2013 in the Kathmandu valley region of Nepal of O3 and CO and contemporaneous measurements of these two gases from Nainital and Pantanagar in India. Hydrocarbon data was collected for a few weeks in December and January during the same period in Bode, Nepal. Extensive analysis of this data is presented contrasting O3 and CO for the winter and spring and episode resulting from biomass burning over western parts of India on the air masses observed over the three sites is presented. The dataset is unique and certainly worthy of discussion in a paper. One serious short coming of the paper is

an absence of model calculations supporting numerous statements in the manuscript. I can't see how this will make sense without supporting model simulations. I will detail some of them below. A second un answered question is what connects these three sites besides them being in approximately near and in the Himalayan foothills? Kathmandu is in a valley and that makes it meteorology fairly unique and extensively influenced by drainage flows, flow through mountain passes and other complex flow situations. Naintal site is at the edge of the mountain ranges and high enough that it can be considered a background site. Pantnagar being lower elevation and in the IGP and is potentially not directly connected to Kathmandu (in terms of transport). I would like to see some discussion why these three sites make a good case for comparing with each other. Finally, the use of HYSPLIT for boundary layer flow reaching Bode in the central of the Kathmandu valley with a 1 degree x 1 degree GDAS fields is probably not a good idea. It would be reasonable to use this method for trajectories reaching mid troposphere. Due to the complex flow conditions here you need much higher resolution flow fields and may be trajectories that reach the top of PBL at Bode rather than the surface. I recommend they try using higher resolution (0.5 degree?) flow fields or better (generated with WRF simulations for example) to increase the confidence in these trajectories.

Specific Comments:

Page 14, Line 1: photo-dissociation of NO3 and NO5 at sunrise Q: Have you measured, No3, N2O5, HNO3 or NOy ever during this experiment. Are there model caluclations that show how much N2O5 can be produced during nighttime? Are there any estimates of PAN produced using models or observations? It is really hard to tell this complicated story using CO and O3. You will need lot more measurements to constrain your story and these are basic measurements for any airuqality study.

Page 14, Line 5 to 10: discussion on ozone mix down Q: Are there any measurements of ozone profiles at Bode? Ozone sondes etc? In its absence running a model may help evaluate these claims. As it stands this is pure speculation and unsupported by

any facts and the discussion is very qualitative.

Page 15, line 5 to 10: Decrease of CO from morning to evening Q: This is again a fairly qualitative description with no supporting data. Later in the discussion it seems like the photochemistry is active during the early morning hours. What are the OH levels at Bode during early morning and late afternoon time periods? A model will help distinguish between chemical and meteorological phenomena.

Page 17, Line 1 to 5: Seasonal variability of CO, decrease from winter to Spring Q: Again, I am not sure how much role chemical loss of CO is important here. Having an idea of OH concentration changes between winter and spring in the valley would be useful. This being a valley the CO emitted could stay trapper for much longer times than other places and hence photochemistry plays a bigger role in CO lifetime.

Page 18, Line 1:10: ozone variation from winter to spring Q: What happens to NO emissions from Winter to Spring? If CO emissions are said to be decreasing will it also not lower NO emissions? Does a decrease in fresh NO at evening hours keep more of the ozone from losses during the night? This would make the ozone issue mostly local.

Page 18, Line 11 – 15: negative correlations between CO and ozone Q: what is the explanation for this negative correlation?

Page 18, Line 20:24: measurements from Nainital and Pantanagar Q: How are these sites connected meteorologically?

Page 20: Line 2-5: No titration discussion Q: This probably is the explanation for the ratios of CO and ozone

Page 21, Line 6: Bode are likely Q: do you mean 'unlikely'?

Figure 11: ozone, co time series Q: what about the CO and ozone peaks in April? Are they also from biomass burning?

---

## Short Comment (SC1) · 1 Sep 2017

Rupakheti et al (2016) reported the ambient concentration of BC, PM, CO and Ozone in Lumbini (regional site of SusKat campaign) during April-June 2013. Two episodes were observed (7-9th April and 3-4th May) during the measurement period which were proved (using HYSPLIT) to be influenced by the air masses travelling over the open burning in NW IGP region. During these events all measured species exhibited peaks. We would like to request the authors to include the findings from our study in the context that enriched concentration were also observed over Lumbini (like Bode) because of

the emission from open burning over NW IGP.

Reference: Rupakheti, D., Adhikary, B., Praveen, P. S., Rupakheti, M., Kang, S., Mahata, K. S., Naja, M., Zhang, Q., Panday, A. K., and Lawrence, M. G.: Pre-monsoon air quality over Lumbini, a world heritage site along the Himalayan foothills, Atmos. Chem. Phys. Discuss., https://doi.org/10.5194/acp-2016-430, in review, 2016 (Accepted for ACP).

Regards Dipesh Rupakheti and Shichang Kang

---

## Referee Comment (RC2) · Anonymous Referee #3 · 9 Feb 2018

The manuscript presents observations of ozone, carbon monoxide and some of the hydrocarbons at Bode in the Kathmandu Valley for a period of 6 months. A correlation analysis and comparison with stations in the northern Indian subcontinent are conducted, and effects of biomass burning are studied. However, manuscript in its present form adds limited new insights into the chemistry and dynamics over this region, and the observations of ozone and carbon monoxide shown here for a period of about 6 months are subset of full year data at same station presented in Mahata et al., 2017. The discussions in the present version are qualitative and general, as elaborated in

following comments

Regional sources are suggested as the driver of springtime ozone enhancement over Bode, however, Fig. 10 clearly shows that during both winter and spring the ozone production is faster over Bode (and Paknajol) as compared to that over IGP stations. For the transport to be the driver, one would expect just an enhancement in the levels with lesser local production, not being evident here.

Springtime enhancement in ozone at Bode appears primarily due to a broader ozone maxima in the noontime which authors attribute to solar radiation (Page 14, lines: 3-5), and higher ozone levels during the nighttime attributed to lower titration with NOx (Page 20, Line 2). Both of these processes are of local origin.

Effect of biomass burning: It would be more appropriate to show a time series of fire counts over potential source region in north India, a running mean of fire counts would be even better as done previously (Kumar et al., JGR, 2011) to classify High and low fire activity periods. Presently, the selected period shows only small enhancements in CO, while larger CO enhancements are seen starting from middle-April.

Page 10, lines.4-5: Why a high-resolution regional-scale model is not used to study transport from such nearby regions of north India to Nepal?

Page 13: l.17-20: It should be useful to substantiate the statements by calculating the variations in ventilation coefficients using measured boundary layer height and wind speed.

Section 3.5: Correlation between ozone and CO: This section needs thorough revision. The anti-correlation is being explained by contrasting seasonality of ozone and CO. Then what is learnt from plotting correlations? Correlating between noontime ozone and CO showing positive relationship is useful. Fig. 7 could be removed.

Stratosphere-to-troposphere transport (STT) could be important in ozone variations over the Himalayas during winter (Phanikumar et al., 2017) as well as spring (Sarangi

et al., 2014). Smaller enhancements in CO but high ozone levels could have some effects of STT. Mountain regions in Nepal also experience such effects. What is the contribution of STT from northern India, or high altitudes in Kathmandu in the ozone enhancements during May event, and in general from Jan to May?

Other comments

Section 2.1.: There should be a description of emission sources, and as paper aims to highlight the differences in the emissions from India and Kathmandu, it should have been substantiated by showing emission maps from recent inventories for some of the chemical tracers.

Too general statements should be removed. Such as Page 11: l.17-18: "The solar radiation. . ...the valley"

Page 11: l.11 "The regional contribution in this regard can not be ruled out". Can it be ruled out in other seasons / or other stations?

With availability of data, there should have been some quantitative discussions. For example: Page 20, l.2-4: "indicating somewhat lesser polluted kind of environment in Bode. . ..However this does not necessarily mean that NOx emissions are lower in Kathmandu valley."

Page 15: l.19: "Methane levels are much higher than the global average". Mention a global average value for the period or from some reference and indicate by how much % it is found to be higher at Bode?

Page 16, l.1-3: ". . .hydrocarbons are much higher than at Nainital. . ..". Mention some % or factor.

Page 21: l.18-21: This discussion is about aerosol forcing, which is not studied in this paper so it can either be moved to introduction or can be skipped.

Page 21: l.22-23: CO enhancement seems small and within the variations, while ozone

enhancements are observed in the following days too.

References

Kumar, R., M. Naja, S. K. Satheesh, N. Ojha, H. Joshi, T. Sarangi, P. Pant, U. C. Dumka, P. Hegde, and S. Venkataramani(2011), Influences of the springtime northern Indian biomass burning over the central Himalayas, J. Geophys. Res., 116, D19302, doi:10.1029/2010JD015509.

Mahata, K. S., Rupakheti, M., Panday, A. K., Bhardwaj, P., Naja, M., Singh, A., Mues, A., Cristofanelli, P., Pudasainee, D., Bonasoni, P., and Lawrence, M. G.: Observation and analysis of spatio-temporal characteristics of surface ozone and carbon monoxide at multiple sites in the Kathmandu Valley, Nepal, Atmos. Chem. Phys. Discuss., https://doi.org/10.5194/acp-2017-709, in review, 2017.

Phanikumar, D. V., K. N. Kumar, S. Bhattacharjee, M. Naja, I. A. Girach, P. R. Nair, and S. Kumari (2017), Unusual enhancement in tropospheric and surface ozone due to orography induced gravity waves, Remote Sensing of Environment, 199, 256-264. https://doi.org/10.1016/j.rse.2017.07.011

Sarangi, T., M. Naja, N. Ojha, R. Kumar, S. Lal, S. Venkataramani, A. Kumar, R. Sagar, and H. C. Chandola (2014), First simultaneous measurements of ozone, CO, and NOy at a high-altitude regional representative site in the central Himalayas, J. Geophys. Res. Atmos., 119, 1592–1611, doi:10.1002/2013JD020631.

---

## Author Comment (AC1) · 17 May 2018

**Variations in surface ozone and carbon monoxide in the Kathmandu Valley and surrounding broader regions during SusKat-ABC field campaign: Role of local and regional sources**

By Piyush Bhardwaj et al., 2017 (ACPD)

We would like to thank the referees for their comments and suggestions and the editor for providing an extended period that allowed us to adequately respond to the comments and improve quality of the manuscript. Here, please find the reviewer's comments in regular black font and our responses in regular blue font. All the changes made in the revised manuscript are indicated in red color. The line numbers in our response refers to the line numbers in the revised manuscript.

**Reviewer-2**

The paper presents data collected during the winter and spring of 2012-2013 in the Kathmandu valley region of Nepal of O3 and CO and contemporaneous measurements of these two gases from Nainital and Pantnagar in India. Hydrocarbon data was collected for a few weeks in December and January during the same period in Bode, Nepal. Extensive analysis of this data is presented contrasting O3 and CO for the winter and spring and episode resulting from biomass burning over western parts of India on the air masses observed over the three sites is presented. The dataset is unique and certainly worthy of discussion in a paper.

Thank you very much for appreciating the extensive analysis and uniqueness of data from three sites that brought-out contrasting features here. Here, we have addressed all your concerns including the analysis of model results.

One serious short coming of the paper is an absence of model calculations supporting numerous statements in the manuscript. I can't see how this will make sense without supporting model simulations. I will detail some of them below.

We have now added model (WRF-Chem) simulations performed over the Kathmandu region. The model output is used to support some of the statements, while other general statements without supporting evidence are removed. Aurelia Lupascu, of the IASS, Potsdam, Germany has kindly shared her model simulations with us and so, she has been added as a co-author in this work. A brief description of the model configuration along with discussions of model results has been added in the revised manuscript.

A second un answered question is what connects these three sites besides them being in approximately near and in the Himalayan foothills? Kathmandu is in a valley and that makes it meteorology fairly unique and extensively influenced by drainage flows, flow through mountain passes and other complex flow situations. Nainital site is at the edge of the mountain ranges and high enough that it can be considered a background site. Pantnagar being lower

elevation and in the IGP and is potentially not directly connected to Kathmandu (in terms of transport). I would like to see some discussion why these three sites make a good case for comparing with each other.

We agree with the reviewer that it is very important point to highlight the importance of considering these three sites together. We tried to mention it in the Introduction of the previous version of the manuscript and we apologize that it was not clear to you. The goal of the SusKat field campaign was not only to perform a detailed characterization of air quality in the Kathmandu Valley but also to understand how the magnitude and variability of air pollution in Kathmandu Valley compares with other sites in the region especially with those located in the cleaner Himalayas and the Indo-Gangetic Plain (IGP). To achieve this goal, Indian institutions also participated in the field campaign by deploying some of their instruments in the Kathmandu Valley and by sharing air quality measurements from their own sites in India. Thus, Nainital (a background site representative of the central Himalayan environment) and Pantnagar (representative of the IGP environment) sites were part of the SusKat field campaign. While most of the previous studies published from the SusKat field campaign focused on analysis of measurements performed in Kathmandu Valley, this study takes a step further to present the regional picture during SusKat and attempts to understand the similarities and differences between the air quality of Kathmandu Valley and the Indian sites, to identify the regional emission sources that are common to these sites. This information has been included in the revised manuscript.

Finally, the use of HYSPLIT for boundary layer flow reaching Bode in the central of the Kathmandu valley with a 1 degree x 1 degree GDAS fields is probably not a good idea. It would be reasonable to use this method for trajectories reaching mid troposphere. Due to the complex flow conditions here you need much higher resolution flow fields and may be trajectories that reach the top of PBL at Bode rather than the surface. I recommend they try using higher resolution (0.5 degree?) flow fields or better (generated with WRF simulations for example) to increase the confidence in these trajectories.

Thank you very much for this comment. Earlier, we were unable to obtain higher resolution wind field data. Now, we have managed to get higher resolution ($0.5^{o}$x$0.5^{o}$) meteorological fields from the Global Data Assimilation System (GDAS) and same are now used to generate the 5-day back-air trajectories over the Bode region. We have now revised this information in Section 2.4 (Satellite data, model and back-air trajectory). The patterns of the back-air trajectory remain more-or-less similar and do not affect conclusions of the study. We have replaced Figure 4 with the new trajectories and text has also been revised in section 3.2. Back-air trajectories calculated with higher resolution wind fields are also shown below for your ready reference.

[Figure]

**Figure 1:** *Five days nine particles HYSPLIT back-air trajectories over Bode region during (a) January, (b) March, (c) May and (d) June. The colored trajectories are of monthly averaged for each nine particles during the respective months.*

Specific Comments:

Page 14, Line 1: photo-dissociation of NO3 and NO5 at sunrise Q: Have you measured, NO3, N2O5, HNO3 or NOy ever during this experiment. Are there model calculations that show how much N2O5 can be produced during nighttime? Are there any estimates of PAN produced using models or observations? It is really hard to tell this complicated story using CO and O3. You will need lot more measurements to constrain your story and these are basic measurements for any air quality study.

Unfortunately, the measurements of $NO_3$, $N_2O_5$, $HNO_3$ or $NO_y$ and PAN were not conducted during the campaign, and lack of systematic observations of these species in South Asia remains a long-standing issue. Therefore, as suggested by the reviewer, we use the model output to gain some process-level understanding of this complex system. The model simulated diurnal variations in NO, $NO_2$, $NO_3$ and $N_2O_5$ during February and May of 2013 are shown in Figure 2 (below). NO mixing ratios are close to zero during the nighttime because it rapidly reacts with $O_3$ to form $NO_2$, which also explains higher $NO_2$ levels during nighttime. $NO_3$ and $N_2O_5$ also show higher levels during nighttime because of the reactions of $NO_2$ with $O_3$, of $NO_2$ with $NO_3$, respectively. The sharp morning increase in NO mixing ratios correlates strongly with the sharp decrease in $NO_3$ and $N_2O_5$ mixing ratios especially during February indicating that photodissociation of $NO_3$ ($\lambda$ <670 nm) and $N_2O_5$ (280< $\lambda$ <380nm) releases NO back to the atmosphere. There is likely some contribution from the $NO_2$ photolysis as well. This figure

has been added as Figure 6 in the revised manuscript and discussion has been added in Section 3.3.

[Figure]

**Figure 2:** *Model simulated average diurnal variations in NO, NO₂, NO₃ and N₂O₅ during February and May 2013.*

Page 14, Line 5 to 10: discussion on ozone mix down Q: Are there any measurements of ozone profiles at Bode? Ozone sondes etc? In its absence running a model may help evaluate these claims. As it stands this is pure speculation and unsupported by any facts and the discussion is very qualitative.

Unfortunately, vertical measurements of ozone are not made at Bode site during this field campaign. Therefore, we have now removed this sentence from the revised manuscript.

Page 15, line 5 to 10: Decrease of CO from morning to evening Q: This is again a fairly qualitative description with no supporting data. Later in the discussion it seems like the photochemistry is active during the early morning hours. What are the OH levels at Bode during early morning and late afternoon time periods? A model will help distinguish between chemical and meteorological phenomena.

We have now added the observations of the boundary layer height and discussion have been revised (section 3.4, page 16) accordingly. OH, observations were not made during the campaign and its direct observations are not existing in the South Asia. But, yes, we agree that CO loss via OH will also contribute and we have modified this accordingly (section 3.4 page 16). We have also added estimates of ventilation coefficient and showed that higher wind speed

leads to lower CO levels

Page 17, Line 1 to 5: Seasonal variability of CO, decrease from winter to Spring Q: Again, I am not sure how much role chemical loss of CO is important here. Having an idea of OH concentration changes between winter and spring in the valley would be useful. This being a valley the CO emitted could stay trapper for much longer times than other places and hence photochemistry plays a bigger role in CO lifetime.

We agree with you and we have revised this part with mention of role of OH chemistry. Model simulated OH levels are found be higher in May, when compared with February. This would suggest great chemical loss of CO in spring (May).

Page 18, Line 1:10: ozone variation from winter to spring Q: What happens to NO emissions from Winter to Spring? If CO emissions are said to be decreasing will it also not lower NO emissions? Does a decrease in fresh NO at evening hours keep more of the ozone from losses during the night? This would make the ozone issue mostly local.

It is difficult to comment on winter to spring changes in CO and $NO_x$ emissions with our dataset because some emission sources (e.g., brick kiln industries and domestic heating) operate only during winter while others (e.g., crop residue burning and forest fires) are more active during the spring. We do not anticipate large variability in other anthropogenic sources such as cooking, traffic and power generation. An accurate characterization of seasonal variability in CO and NOx emissions over this region will require the development of an emission inventory considering the temporal variability in all the emission sources, which is beyond the scope of present study. The increase in biomass burning activity is supported by a detailed analysis of MODIS active fire locations and this discussion has been extensively revised in section 3.7 (Influences of springtime northern Indian biomass burning). Please see our responses to Reviewer#3). We also analyzed OMI and GOME-2 retrieved tropospheric column $NO_2$ (cloud screened 30%) over the Bode region (27-28ºN, 85-86ºE) to understand variability in the tropospheric column burden of $NO_2$. Both the satellites show similar levels (below figure 3) in winter and spring (except somewhat higher levels in early April which we feel are due to biomass burning in nearby regions).

In addition to changes in emissions, we envisage that variations in the PBL height between winter and summer can significantly affect the surface concentration of different air pollution. The observations of the boundary layer height were made during the campaign using a ceilometer and those clearly show higher boundary layer height during spring (pre-monsoon) i.e. March, April and May (Figure 3 above; please also see Figure 4 from Mues et al., 2017). Similar increase in PBL height is seen in the model results. If we assume that emissions are constant from winter to spring, then increase in the PBL height will lead to lower mixing ratios of species such as CO and NOx by allowing the emissions to mix into a larger volume compared to the winter.

We also agree with the reviewer that lower NO levels during evening hours could reduce the ozone loss. The model simulated NO levels are lower in spring ((Figure 6 of the revised

manuscript) that will lead to reduction in nighttime ozone loss in spring. This discussion has been included in the revised manuscript.

[Figure]

**Figure 3:** *OMI tropospheric column NO₂ (cloud screened 30%) and GOME-2 tropospheric column NO₂ over Bode.*

Page 18, Line 11 – 15: negative correlations between CO and ozone Q: what is the explanation for this negative correlation?

Negative correlation during nighttime and early morning time is a manifestation of ozone titration by $NO_x$ and the lower boundary layer height. We have now revised the text accordingly (section 3.5).

Page 18, Line 20:24: measurements from Nainital and Pantanagar Q: How are these sites connected meteorologically?

Nainital and Pantnagar are meteorologically disconnected during the nighttime and early morning hours because the boundary layer height remains below the altitude of Nainital. However, the boundary layer height is higher the altitude of Nainital during afternoon and thus are connected meteorologically. In the afternoon, Pantnagar acts as a representative of emission sources affecting Nainital. We have now added the discussion in section 2.1 (Observation site) in the revised manuscript. Details of both sites have been provided in previous publications (Kumar et al., 2010, Ojha et al., 2012; Sarangi et al, 2014; Naja et al., 2014; Joshi et al., 2016).

Page 20: Line 2-5: No titration discussion Q: This probably is the explanation for the ratios of CO and ozone

We agree. We have added this in the response of previous comments also.

Page 21, Line 6: Bode are likely Q: do you mean 'unlikely'?

Sorry for the mistake. We mean, 'unlikely' and now the statement is revised accordingly.

Figure 11: ozone, co time series Q: what about the CO and ozone peaks in April? Are they also

from biomass burning?

Yes, they are also due to biomass burning. Now, we have further improved the biomass burning analysis (as suggested by reviewer-3) over the region to identify the role of biomass burning in the CO and $O_3$ peaks during April. These peaks were identified to be highly correlated to the high biomass burning activity which occurred in the nearby regions surrounding the Kathmandu valley. The HYPLIT calculated back-air trajectories also show transport from the active fire regions to Kathmandu Valley during this period. The air-masses during this period were found to be circulating in the region. A similar episode was also detected during the early May over the region where influences of northern Indian biomass burning was investigated. The entire section (3.7) regarding biomass burning is rewritten to answer this and other similar questions.

**Reviewer-3**

The manuscript presents observations of ozone, carbon monoxide and some of the hydrocarbons at Bode in the Kathmandu Valley for a period of 6 months. A correlation analysis and comparison with stations in the northern Indian subcontinent are conducted, and effects of biomass burning are studied. However, manuscript in its present form adds limited new insights into the chemistry and dynamics over this region, and the observations of ozone and carbon monoxide shown here for a period of about 6 months are subset of full year data at same station presented in Mahata et al., 2017. The discussions in the present version are qualitative and general, as elaborated in following comments

We thank the reviewer for a thorough review of our manuscript. However, we strongly disagree with the reviewer's assessment that this study simply presents a subset of yearlong ozone and CO observations presented in Mahata et al. (2017). This study differs from Mahata et al. (2017) particularly in the sense that Mahata et al. (2017) focuses on observations conducted in and around Kathmandu Valley while this study for the first time provides a broader regional picture by complementing the observations from Kathmandu Valley with simultaneous observations from a high-altitude background site located in the central Himalayas (Nainital) and a semi-urban site representative of the chemical environment of the Indo-Gangetic Plain. Since most of the previous studies have already provided key information about air quality in Kathmandu Valley, this study takes a step further to present the regional picture during SusKat and attempts to understand the similarities and differences between the air quality of Kathmandu Valley and the Indian sites, to identify the regional emission sources that are common to these sites. Furthermore, the revised manuscript complements the observations with model simulations following suggestions from both the reviewers and also discusses the role of biomass burning in greater detail. Below, we respond to all the reviewer comments one by one with the comments reproduced in black font and our response in blue font.

Regional sources are suggested as the driver of springtime ozone enhancement over Bode, however, Fig. 10 clearly shows that during both winter and spring the ozone production is faster over Bode (and Paknajol) as compared to that over IGP stations. For the transport to be the driver, one would expect just an enhancement in the levels with lesser local production, not being evident here.

We agree with the reviewer that in situ ozone production resulting from photooxidation of precursor gases is the main source of ozone at Bode as reflected by faster ozone production rates in Figure 10 (now Figure 12). However, we wanted to point out that regional transport adds ozone on top of the already higher in situ photochemical ozone production at Bode especially during springtime. An increase in regional-scale $O_3$ levels during the spring season has also been reported in several previous studies (e.g., Kumar et al., 2010, Ojha et al., 2012; Sarangi et al., 2014; Putero et al., 2015), and it seems to be playing an important role at Bode as well. We have revised the statements in the manuscript to reflect this discussion.

Springtime enhancement in ozone at Bode appears primarily due to a broader ozone maxima in the noontime which authors attribute to solar radiation (Page 14, lines: 3-5), and higher ozone levels during the nighttime attributed to lower titration with NOx (Page 20, Line 2). Both of these processes are of local origin.

The reviewer's question is not clear here, but we believe that the reviewer is advocating that local ozone production is the major contributor at Bode. We agree with the reviewer's opinion here.

Effect of biomass burning: It would be more appropriate to show a time series of fire counts over potential source region in north India, a running mean of fire counts would be even better as done previously (Kumar et al., JGR, 2011) to classify High and low fire activity periods. Presently, the selected period shows only small enhancements in CO, while larger CO enhancements are seen starting from middle-April.

Thanks for this comment. We have now revised this analysis completely as suggested. We have classified the observations in high and low fire activity periods, and quantified fire induced enhancements in $O_3$ and CO. During the spring season, two distinct peaks in $O_3$ and CO over Bode have been studied separately using the same methodology. To improve the quality of discussions, emissions from a high-resolution biomass burning emissions inventory and satellite retrievals are also used. We have now revised the section 3.7 and new figures and tables have also been added in the revised manuscript. The figures and table are reproduced below from the revised manuscript for your ready reference.

[Figure]

**Figure 4:** *Top Left: time series of MODIS daily fire counts (red bar), 3-day running mean (black lines), median fire counts (brown line) for the fire period (3 April31 May 2013). Total biomass burning emissions (orange line), crop residue burning emissions (dark yellow line) and forest fire emissions (dark green) over 1ºx1º grid box around bode (27-28ºN, 85-86ºE) are also shown. Top right: average biomass burning emissions for two fire activity periods over Bode region. Center: time series of surface ozone mixing ratios at Bode and Nainital (line plot-Left) and average ozone mixing ratios during two fire periods (as bar plot-Right), respectively. Bottom: time series of surface CO mixing ratios at Bode and Nainital (line plot-Left) and average CO mixing ratios during two fire periods (as bar plot-Right), respectively. The two fire events in April and May are also shown highlighted (in violet boxes).*

[Figure]

***Figure 5:*** *Spatial distribution of MODIS fire counts during April (red dots) and May (orange dots) 2013. Underneath is daily HYSPLIT backward trajectories during April (blue lines) and May (green lines) initiating from Bode (black star). The two black boxes indicate two hotspots of fire counts during this period.*

[Figure]

***Figure 6:*** *Top (1a-3a): Spatial distribution of MODIS fire counts over Northern Indian subcontinent during three periods (left to right). The black boxes represent two fire hotspots (refer Fig. S1) over the region with fire counts shown as different colors during the three periods. Center (1b-3c): spatial distribution of AIRS CO mixing ratio at 925hPa (1b-3b) and 850 hPa (1c-3c) during three periods (left to right). Center (1d-3d): Average biomass burning emissions over Bode region (27-8ºE, 85-86ºE) using GFED v4.0 inventory during the three periods (left to right). Bottom Panels (1e-3f): changes in average surface mixing ratios of O$_3$ (1e-3e) and CO (1f-3f) at different sites during the three periods (left to right).*

[Figure]

**Figure 7:** *Top (1a-2b): Spatial distribution of MODIS fire counts over Northern Indian subcontinent during two periods (left to right). The black boxes represent two fire hotspots (refer Fig. S1) over the region with fire counts shown as different colors during different periods. Center (1b-2c): spatial distribution of AIRS CO mixing ratio at 925hPa (1b-2b) and 850 hPa (1c-2c) during two periods (left to right). Center (1d-2d): Average biomass burning emissions over Bode region (27-8ᵒE, 85-86ᵒE) using GFED v4.0 inventory during the two periods (left to right). Bottom Panels (1e-2f): changes in average surface mixing ratios of O₃ (1e-2e) and CO (1f-2f) at different sites during the two periods (left to right).*

**Table 4:** The average O$_3$ and CO mixing ratios at Bode, Nainital, Pantnagar with GFED average biomass burning emissions over Kathmandu region during different periods.

| Fire Periods | Fire Count | Ozone (ppbv) | | CO (ppbv) | | Avg. Biomass burning emissions (Tg/day) | | |
|---|---|---|---|---|---|---|---|---|
| | | Bode | NTL | Bode | PNT | Total | Crops | Forest |
| LFAP (Mar 1-Mar 31) | 5 | 37.4 | 45.2 | 705 | 455 | 8.96 | 2.29 | 5.87 |
| HFAP (Apr-May) | 70 | 43.7 | 63.9 | 504 | 374 | 63.70 | 42.70 | 19.94 |
| | | | | | | | | |
| Mar (20-25) | 3 | 34.8 | 46.1 | 693 | 401 | 7.17 | 2.37 | 3.67 |
| Apr (3-6) | 18 | 46.7 | 49.2 | 797 | 250 | 57.74 | 7.48 | 46.46 |
| Apr (11-16) | 27 | 54.0 | 57.9 | 762 | 265 | 52.38 | 7.03 | 44.16 |
| | | | | | | | | |
| Apr (26-30) | 26 | 33.9 | 45.0 | 505 | 313 | 22.10 | 14.60 | 6.78 |
| May (2-6) | 116 | 58.8 | 74.8 | 625 | 495 | 128.40 | 97.58 | 29.72 |

Page 10, lines.4-5: Why a high-resolution regional-scale model is not used to study transport from such nearby regions of north India to Nepal?

Following the reviewer suggestion, we have now added model (WRF-Chem) simulations results and the discussion has been revised extensively. We have also included figures comparing the model results with the observations.

Page 13: l.17-20: It should be useful to substantiate the statements by calculating the variations in ventilation coefficients using measured boundary layer height and wind speed.

We have now added observations of the boundary layer height. Information on ventilation coefficients have also been added. The figure of ventilation coefficient is shown below for the ready reference.

[Figure]

*Figure 8: The average diurnal variations in ozone, CO, median MLH and Ventilation coefficient during winter (top) and Spring (bottom).*

Section 3.5: Correlation between ozone and CO: This section needs thorough revision. The anti-correlation is being explained by contrasting seasonality of ozone and CO. Then what is learnt from plotting correlations? Correlating between noontime ozone and CO showing positive relationship is useful. Fig. 7 could be removed.

We have now revised this section and added discussion on the correlation. We would like to retain the figure 7 (now fig 9) as it provides very useful information on changes in correlation with months. It very clearly shows decrease in correlation from winter to spring.

Stratosphere-to-troposphere transport (STT) could be important in ozone variations over the Himalayas during winter (Phanikumar et al., 2017) as well as spring (Sarangi et al., 2014). Smaller enhancements in CO but high ozone levels could have some effects of STT. Mountain regions in Nepal also experience such effects. What is the contribution of STT from northern India, or high altitudes in Kathmandu in the ozone enhancements during May event, and in general from Jan to May?

We have added a complete new section (3.8) to address the stratosphere troposphere exchange in the manuscript. In this section the vertical variations in potential vorticity, CO, relative humidity and ozone were discussed to check the role of stratosphere troposphere transport (STT) from January to May 2013. The updated figures are added below.

[Figure]

***Figure 9:*** *Vertical distribution of ertel potential vorticity (EPV) calculated by MERRA v2 reanalysis (top), AIRS retrieved ozone mixing ratios (centre), CO mixing ratios (center) and*

*relative humidity (RH) during Jan-May 2013.*

[Figure]

***Figure 10:*** *Vertical profiles of EPV, ozone, CO and RH during April (Top Panels) and May (Bottom Panels) biomass burning episodes.*

Other comments

Section 2.1.: There should be a description of emission sources, and as paper aims to highlight the differences in the emissions from India and Kathmandu, it should have been substantiated by showing emission maps from recent inventories for some of the chemical tracers.

We appreciate this suggestion and we have added emission maps for CO and NOx (in the supplementary material) from EDGAR-HTAP emission inventory. Details of emission inventories are also available at http://edgar.jrc.ec.europa.eu/htap_v2).

Too general statements should be removed. Such as Page 11: l.17-18: "The solar radiation. . ...the valley"

These lines were removed from the manuscript.

Page 11: l.11 "The regional contribution in this regard cannot be ruled out". Can it be ruled out in other seasons / or other stations?

The high levels of air pollutants in Kathmandu Valley have been suggested to be primarily influenced by local emissions from the valley. In this study, we also observed that except for a few cases the air quality at Kathmandu is constrained by in-situ chemistry and emissions. Therefore, we wanted to emphasize that regional contribution can be important occasionally. Nevertheless, we have now removed this sentence.

With availability of data, there should have been some quantitative discussions. For example: Page 20, l.2-4: "indicating somewhat lesser polluted kind of environment in Bode. . ...However this does not necessarily mean that NOx emissions are lower in Kathmandu valley."

We have now added the information on the previous observations where NO levels are reported to be as high as 60 ppbv. More discussions on this have also been added in section 3.3.

Page 15: l.19: "Methane levels are much higher than the global average". Mention a global average value for the period or from some reference and indicate by how much % it is found to be higher at Bode?

We have now provided the needed information (section 3.4). Nevertheless, it is quite obvious that methane levels more than 2 ppmv could be considered as higher levels.

Page 16, l.1-3: ". . .hydrocarbons are much higher than at Nainital. . ..". Mention some % or factor.

Yes, now we have provided this information. Nevertheless, values were already given in table 3.

Page 21: l.18-21: This discussion is about aerosol forcing, which is not studied in this paper so it can either be moved to introduction or can be skipped.

These lines are removed from the manuscript.

Page 21: l.22-23: CO enhancement seems small and within the variations, while ozone enhancements are observed in the following days too.

We have now revised the discussion. Some of these ozone enhancements are discussed in the biomass burning analysis section (3.7) and other sections.

---

## Author Comment (AC2) · 17 May 2018

Thank you very much, we have included the mentioned reference.